# Phase-separated CCER1 coordinates the histone-to-protamine transition and male fertility

Dongdong Qin[1,7], Yayun Gu[1,2,7], Yu Zhang[3,4,5,6,7], Shu Wang[1,7], Tao Jiang[1,2], Yao Wang[3,4], Cheng Wang [1,2], Chang Chen[1], Tao Zhang[1], Weiya Xu [1], Hanben Wang[1], Ke Zhang[3,4], Liangjun Hu[3,4], Lufan Li[1], Wei Xie [3,4] ✉, Xin Wu [1] ✉ & Zhibin Hu [1,2] ✉

Idiopathic fertility disorders are associated with mutations in various genes. Here, we report that coiled-coil glutamate-rich protein 1 (CCER1), a germline-specific and intrinsically disordered protein (IDP), mediates postmeiotic spermatid differentiation. In contrast, CCER1 deficiency results in defective sperm chromatin compaction and infertility in mice. CCER1 increases transition protein (*Tnp1/2*) and protamine (*Prm1/2*) transcription and mediates multiple histone epigenetic modifications during the histone-to-protamine (HTP) transition. Immiscible with heterochromatin in the nucleus, CCER1 self-assembles into a polymer droplet and forms a liquid-liquid phase-separated condensate in the nucleus. Notably, we identified loss-of-function (LoF) variants of human *CCER1* (h*CCER1*) in five patients with nonobstructive azoospermia (NOA) that were absent in 2713 fertile controls. The mutants led to premature termination or frameshift in *CCER1* translation, and disrupted condensates in vitro. In conclusion, we propose that nuclear CCER1 is a phase-separated condensate that links histone epigenetic modifications, HTP transitions, chromatin condensation, and male fertility.

Genomic DNA is compacted into nuclear protein assemblies in the nuclei of eukaryotic cells within the nucleosome. Spermatogenesis is one of the most complex and continuous cellular differentiation processes and is characterized by extensive reprogramming of chromatin organization and structure[1]. In contrast to somatic differentiation, the histone-to-protamine (HTP) transition during spermatogenesis is essential for the entire genome to be packaged into the highly concentrated sperm nucleus. Most core histones are initially replaced by testes specific histones, and then transition proteins, followed by protamine proteins, which promote chromatin

compaction and in turn lead to chromatin structural remodeling[2]. In addition to the factors involved in this lineage-specific developmental program, epigenetic regulation is key to the HTP transition. Covalent conjugation of different posttranslational modifications of histones leads to dramatic changes in chromatin conformation, nucleosome stability, and/or histone-DNA interactions during the HTP transition. In general, ubiquitination on testis histone H2 variants promotes histone removal[3], methylation on testis histone H3 regulates transition protein (*Tnp1/2*) and protamine (*Prm1/2*) gene expression[4], whereas the acetylation of histone H4 is essential for

[1]State Key Laboratory of Reproductive Medicine and Offspring Health, Nanjing Medical University, Nanjing, Jiangsu 210029, China. [2]School of Public Health, Center for Global Health, Nanjing Medical University, Nanjing, Jiangsu 211100, China. [3]Center for Stem Cell Biology and Regenerative Medicine, MOE Key Laboratory of Bioinformatics, School of Life Sciences, Tsinghua University, Beijing, China. [4]Tsinghua-Peking Center for Life Sciences, Beijing, China. [5]Obstetrics and Gynecology Hospital, Institute of Reproduction and Development, Fudan University, Shanghai, China. [6]Research Units of Embryo Original Diseases, Chinese Academy of Medical Sciences, Shanghai, China. [7]These authors contributed equally: Dongdong Qin, Yayun Gu, Yu Zhang, Shu Wang. ✉e-mail: xiewei121@tsinghua.edu.cn; xinwu@njmu.edu.cn; zhibin_hu@njmu.edu.cn

destabilization and remodeling of nucleosomes and subsequent incorporation of *Tnps* and/or *Prms*[5].

Liquid-liquid phase separation (LLPS) is a fundamental mechanism for organizing the contents of living cells[6]. Through multivalent interactions, LLPS drives the assembly of various protein aggregates and the formation of membrane-less organelles in cells. LLPS is generally mediated by molecules with intrinsically disordered proteins/ regions (IDP/IDR) and is associated with their low-complexity sequences and prion-like domains. Within the nucleus, phase separation and phase transitions are of particular interest because nuclear condensates must interact with chromatin to control its organization and gene expression. For example, nuclear IDR-driven condensates preferentially form in regions of low chromatin density, where they act as mechanical chromatin filters, excluding untargeted regions of adjacent genomes and reorganizing the genome[7,8]. There is increasing evidence to suggest that nuclear membrane-less organelles such as Cajal bodies, nucleoli, and speckles influence the chromatin structure through LLPS and that LLPS plays critical roles in diverse structures, such as by affecting postsynaptic density, the synaptic complex, and the mitotic spindle[9].

Recently, the fragile X–related (FXR) protein family member FXR1 was identified in cell polysome fractions, which suggest that FXR1 plays a key role in the translation activation of stored mRNAs in mouse spermatids and male fertility in mice through LLPS[10]. Although many studies have described the importance of phase separation and shown that functional imbalances in cellular LLPS condensates orchestrate the assembly of various physiological structures and pathological transformations, the biological evidence of these functions, in particular, germ-cell specific LLPS stories, in terms of their development, has not been discovered; however, these functions are essential for passing genetic information to the next generation. In the present study, we show that nuclear CCER1 (coiled-coil glutamate-rich protein 1), a germline-specific regulator, mediates histone epigenetic modification and chromatin condensation as a phase-separated condensate. The *Ccer1* gene is located on mouse chromosome 10, and the human homologous gene of *CCER1* is located on chromosome 12. Both mouse *Ccer1* and human homologue are single-exon genes. Information from the protein UniProt database showed that the sequence of CCER1 is rich in glutamic acids, while the human sequence contains two coiled-coil domains and the mouse contains one coiled-coil domain. Currently, the *Ccer1* gene lacks any functional study. Importantly, we also show the mutations in human *CCER1* gene link spermatogenesis and male infertility in the population, which is a major issue in human health.

## Results

### Identification of h*CCER1* mutation in patients with azoospermia

Idiopathic infertility is often associated with mutations in genes[11,12], we screened for potential mutations in the gene coding regions in a cohort of 620 patients with NOA and found *CCER1* mutations can be pathogenic to human spermatogenesis. All patients underwent semen analyses on at least three occasions, and those with a history of orchitis, obstruction of the vas deferens, or endocrine disorders were excluded. Sanger sequencing (Supplementary Fig. 1a, b) of the coding region in the *CCER1* gene showed three unique (MAF = 0 in gnomAD populations) loss-of-function variants (c.157 C > T; c.358_388del, 31 bp; c.534 G > A) were identified in five patients with NOA but absent in 2713 fertile controls and were associated with an increased risk of NOA (Fig. 1a, b, $P_{Combined} = 4.551 \times 10^{-7}$). Among these patients, three patients with NOA carried a frameshift mutation (c.358_388del, 31 bp; p.Cys120fs), and two carried stop-gain mutations (c.157 C > T, p.Arg53*; c.534 G > A, p.Trp178*), which may have led to premature termination of translation and loss of CCER1 function (Fig. 1c). Next, we transfected the full-length wild-type and mutant cDNA constructs of CCER1 to HEK293T cells. As a result, we found the degradation of p.Cys120fs

mutants, and truncation of p.Arg53* and p.Trp178* mutants (Fig. 1d). The results suggested that *CCER1* loss-of-function variants might be pathogenic in patients with NOA.

### Germline-specific expression of *Ccer1* is cis-regulated by CpG islands

Because CCER1 studies have not been widely reported in the literature, we first generated an antibody (recognizing the C-terminus of CCER1 at aa 179–403) to analyze the expression of CCER1 in mice. The spatiotemporal distribution showed that the CCER1 protein was expressed only in mouse testes (Fig. 2a) and was significantly elevated starting at P28 and continuing into adulthood (Fig. 2b). CCER1 signal was evident in the nuclei of round-to-elongated spermatids at stages II−X in seminiferous tubules (Fig. 2c), corresponding to steps 2–10 of the 16-step spermatid development process (Fig. 2d). Notably, we observed that CCER1 signal in the testis was consistently absent in intense DAPI-staining and H3K9me3+-labeled regions, reflecting a clear boundary between CCER1 signals and H3K9me3+ heterochromatin in all spermatids (Fig. 2e), and the two regions were clearly immiscible with each other.

Since methylation levels of gene CpG islands (CGIs) often correlate with their tissue-specific expression[13], and *Ccer1* appears to be testis-specific; therefore, we investigated that the CGI methylation level of *Ccer1* in the testes. Using in silico sequence analysis, we found that both human and mouse *CCER1* contain a single exon with successive CGIs located within the gene body, including the 5'UTR and CDS regions, except for the first CGI in mouse *Ccer1*, in which the upstream 88 bp overlap with the transcription start site (TSS), indicating a biased CGI distribution in the *Ccer1* gene (Fig. 2f). Next, we investigated whether *Ccer1* expression in the germline is regulated by CGI in testes. We first examined the CpG sites of the c.404-c.741 region in the mouse *Ccer1* gene body, which is the sequence homologous to the h*CCER1* third CGI sequence, and approximately covered the fourth CGI sequence (c.381-c.557), which is the longest CGI in the mouse gene. Then, we found that the CGI of *Ccer1* was demethylated in mid- or late germ cells during spermatogenesis in the mouse testes but not in the spermatogonia or brain tissue (Fig. 2g). Moreover, we determined the average methylation level of all four CGIs in h*CCER1* and found that the CGIs in h*CCER1* were highly methylated in human tissues in addition to testes (Fig. 2h). Next, we generated a dual-luciferase reporter using a CpG-free reporter vector that was unaffected by DNA methylation[14] and cloned the largest CGI fragment in h*CCER1* (CGI3, 292 bp) into the luciferase sequence upstream of the reporter (Fig. 2i). Notably, higher luciferase expression was found in these plasmids than in CpG-null islands. In contrast, CpG islands were treated with methyltransferase *M.SssI* in vitro, and a decrease in luciferase expression was observed (Fig. 2j, mCpG versus CpG). Taken together, these data suggest that the CGI in *Ccer1* plays a cis-regulatory role and that testis-specific expression of *Ccer1* requires demethylation of the CGIs.

### Deletion of *Ccer1* leads to male infertility in mice

Next, we applied CRISPR–Cas9-mediated gene targeting to generate mutant mice to explore the function of CCER1 (Supplementary Fig. 2a). Three mouse lines with *Ccer1* frameshift mutations were obtained, including a 58-bp deletion, a 29-bp deletion, and an 8-bp deletion in the coding sequence, all of which produced a premature stop codon (hereafter referred to as *Ccer1*−/−). Generations of *Ccer1* mutants were successfully generated from founder lines with indels or deletions of *Ccer1* alleles (Fig. 3a) and the progeny mice we used from each line were all further bred for at least five generations. The *Ccer1*−/− mice showed testicular size and body/testicular weight ratio comparable to those of their wild-type littermates (*Ccer1*+/+, Fig. 3a, b and Supplementary Fig. 2b), and the wild-type and heterozygous mice of littermate controls have no fertility problems; however, in the mating experiment, none of the males from the three lines (−58 bp, −29 bp,

a

| Gene | Variant coordinates | cDNA variation | Amino-acid variation | Maf$_{case}$[a] | Maf$_{gnomAD}$[a] | Homo/ Het | P |
|------|---------------------|----------------|----------------------|----------|-----------|-----------|---|
| CCER1 | Chr12：91348363 | c.157C>T | p.Arg53* | 0.0016 | 0 | Het | 0.054 |
| | Chr12：91348132 | c.358 -388del | p.Cys120fs | 0.0048 | 0 | Het | 1.573x10$^{-4}$ |
| | Chr12：91347986 | c.534G>A | p.Trp178* | 0.0016 | 0 | Het | 0.054 |
| Combined | | | | | | | 4.551x10$^{-7}$ |

b

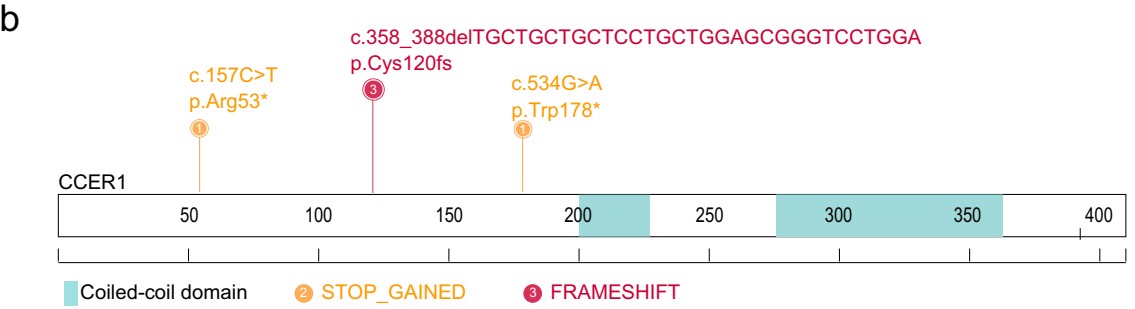

c

d

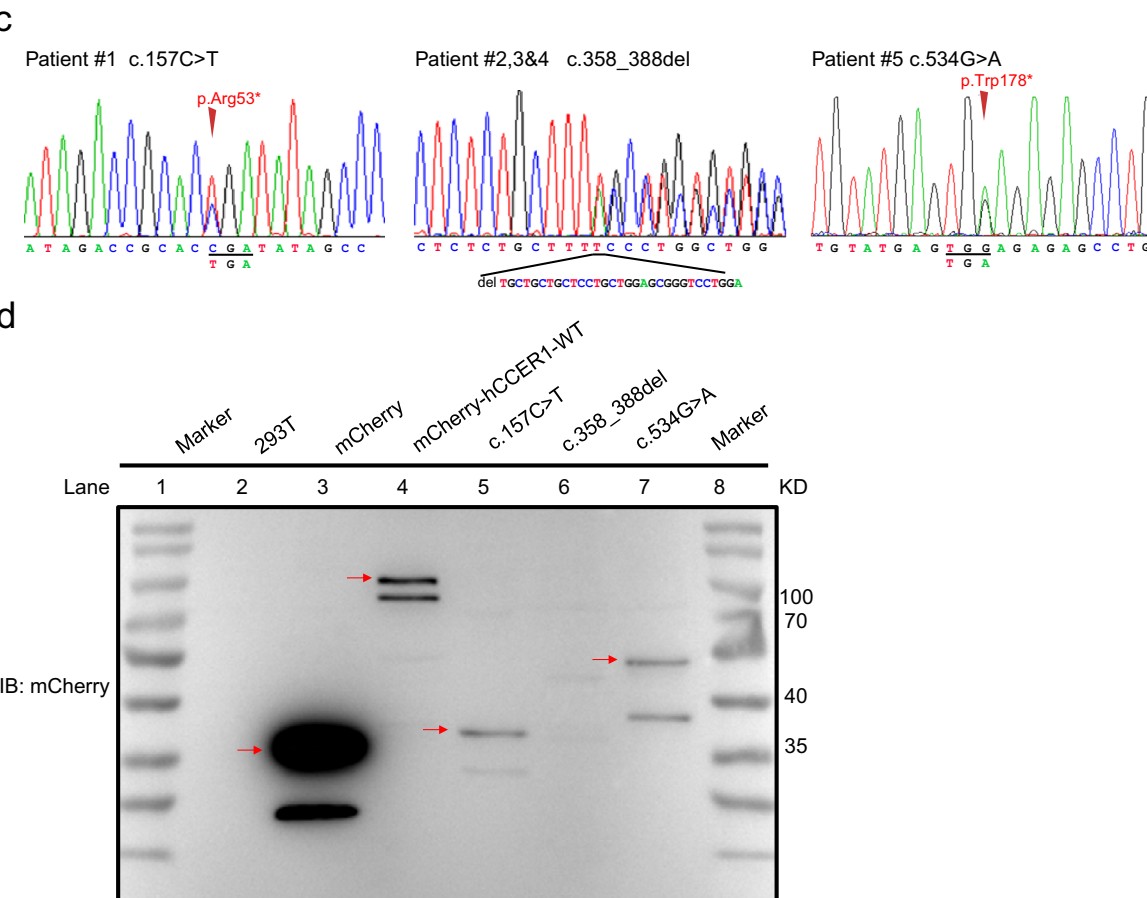

−8 bp) of *Ccer1* mutant mice produced offspring (Fig. 3c), although *Ccer1*$^{-/-}$ females were fertile (*Ccer1* mutant females were used to generate offspring for over 6 months). We selected the 58-bp deletion mouse line for further analysis of pathological changes. Notably, compared to those in *Ccer1*$^{+/+}$ mice, elongating spermatids starting at step 10 in *Ccer1*$^{-/-}$ mice showed malformed spermatids, e.g., the nucleus was overextended, and the structural integrity (hooked head and dorsal angle) was lost. Moreover, the acrosome did not normally extend when its dorsal and ventral surface were lost; therefore, all the elongating spermatids in the following steps (11–16) were globally malformed, and elongated spermatids that should not be found at steps 9 and 10 were still found at stages IX-X in the seminiferous

**Fig. 1 | Identification of unique and deleterious *CCER1* mutations in patients with NOA. a** Association of *CCER1* LOF mutations with the risk of NOA based on 620 patients and 10,847 controls in the gnomAD database. *P* value significance was analyzed by Fisher's exact test ($P_{Combined} = 4.551 \times 10^{-7}$). **b** Diagrammatic representation of the CCER1 protein with known protein domains indicated. The orange mutations represent stop gain mutations, and the red mutations represent frameshift deletions. **c** Chromatogram of the sequences of the *CCER1* coding region for the three abovementioned LOF mutations in patients with NOA. **d** Western blot

to investigate the effects of human mutations on the CCER1 protein levels. Red arrows indicate the predicted bands. From left to right: protein marker (lane 1 & 8), HEK293T cell lysis control (lane 2), HEK293T cells transfected with the mCherry plasmid (lane 3, 35 KD), mCherry-hCCER1-WT (lane 4, approximately 100–110 KD), mCherry-hCCER1-c.157 C > T (lane 5, 35–40 KD truncated protein), mCherry-hCCER1-c.358_388del (lane 6, no truncated protein) and mCherry-hCCER1-c.534 G > A plasmids (lane 7, 40–55 KD), respectively. The samples derive from the same experiment and that blots were processed in parallel.

epithelium (Fig. 3d, e and Supplementary Fig. 2f). In contrast, all types of germ cells were present and appeared normal in the epithelium before step 9 in the spermatids of the testes of the *Ccer1*[−/−] mice compared to the *Ccer1*[+/+] mice (Fig. 3d and Supplementary Fig. 2h). Further examination of the morphology of the seminiferous epithelium in mouse testis by histology revealed delayed sperm release in the testis tubules (Fig. 3e and Supplementary Fig. 2f) and extensively malformed spermatozoa in the epididymis of the *Ccer1*[−/−] mice compared to the *Ccer1*[+/+] mice (Fig. 3f, g). Using a computer-assisted semen analyzer (CASA), we found that the sperm count, percentage of motile sperm, and percentage of forward motile sperm were significantly reduced in the epididymis of *Ccer1*[−/−] mice (Supplementary Fig. 2c–e), which was consistent with the pathological findings in the epididymis (Supplementary Fig. 2g). To further examine *Ccer1*[−/−] sperm function, we performed in vitro fertilization (IVF) assays. Compared with normal sperm, *Ccer1* mutant sperm were rarely able to fertilize oocytes (7.57% for *Ccer1*[−/−] and 88.20% for *Ccer1*[+/+]) (Supplementary Fig. 3a, b).

A scanning electron microscopy (SEM) analysis further showed that spermatozoa in the *Ccer1*[−/−] testes were very abnormal, and these observations of which were consistent with those of the histology analysis (Supplementary Fig. 4a); however, sperm flagella appeared normal, and typical "9 + 2" microtubule structures in the tails were integrated (Supplementary Fig. 4b). Thus, the loss of CCER1 affects the development of spermatids during spermiogenesis (the process of spermatid development) and causes male infertility in mice.

**Loss of CCER1 affects sperm nuclear condensation and the 3D chromatin structure**

Next, we applied transmission electron microscopy (TEM) to further dissect the defects in the sperm from *Ccer1*[−/−] mice. We found that most sperm heads in the *Ccer1*[−/−] epididymis were less condensed than those in the *Ccer1*[+/+] controls and grey intensity ratio of *Ccer1*[−/−] were decreased (Fig. 4a, b), and the acrosome was separated from the nucleus. These observations suggest that sperm chromatin compaction produced by the *Ccer1*[−/−] mice was defective.

To confirm the condensate state in the *Ccer1*[−/−] mutants, we took advantage of in situ Hi-C to illustrate the chromatin state in both wild-type and mutant sperm at the molecular level[15,16] (Supplementary Fig. 5c, d). According to the Hi-C results, the higher-order chromatin structure was altered in the *Ccer1*[−/−] sperm compared to the wild-type control sperm (Fig. 4c–g). The distal interaction gradually decreased in the mutant cells, as shown in an interaction heatmap (Fig. 4c). We confirmed this observation by P(s) curve analysis, which displayed the chromatin contact probability relative to the genomic distance (Fig. 4d). The P(s) curve clearly showed decreased interaction frequency in distal regions, and this result was highly reproducible, indicating different principles of chromatin folding. Moreover, compared to that in WT cells, the compartment was slightly blurred in the *Ccer1*[−/−] sperm (Fig. 4e). At a finer resolution, the dynamics of TADs (topologically associating domains) was analyzed. Both the average TAD interaction frequency and the insulation score results around TADs revealed that the TAD dynamics were weakened in *Ccer1*[−/−] cells (Fig. 4f, g). To summarize, these results support the idea that chromatin in *Ccer1* mutant sperm underwent condensation at a lower rate, and that CCER1 was important for the 3D chromatin organization in

sperm. Moreover, these data were consistent with our observations made via TEM.

**Loss of CCER1 reduces the transcript levels of *Tnp1/2* and *Prm1/2***

CCER1 is present in the nuclear euchromatic region during spermatogenesis, where transcription is thought to be active (although global transcription ceases in haploid germ cells); therefore, we performed RNA-seq with adult *Ccer1*[+/+] and *Ccer1*[−/−] mouse testes. The comparison of mRNA expression profiles revealed 110 upregulated and 72 downregulated genes, representing 0.82% of all transcripts (182/22259 transcripts) that were significantly up- or downregulated (as shown in the pie chart in Supplementary Fig 4c; *P* < 0.05, fold-change>1.5; three independent samples; GEO database, accession no. GSE212733). *Ccer1* deficiency led to a somewhat limited alteration in global gene expression in testes, and CCER1 both activated and suppressed gene expression in round spermatids. Intriguingly, we noted that the most important genes involved in spermatid development, including transition proteins 1 and 2 (*Tnp1/2*) and protamine 1 and 2 (*Prm1/2*), were the most downregulated transcripts (Supplementary Fig. 4d) while the levels of gene transcripts marking various cell-development stages, including *Akap3*, *Tssk6*, *Crem*, *Spaca9*, and *Odf3*, were likely to remain intact (Supplementary Fig. 4e), consistent with the western blot analysis (Fig. 4h). We also performed western blotting and found a significant decrease in TNP1, TNP2, PRM1, and PRM2 protein expression in the *Ccer1*[−/−] testes (Fig. 4i, j). Moreover, a further decrease in the fluorescence intensity of TNP1 and PRM2 was clear in the *Ccer1*[−/−] mouse testes but not in the *Ccer1*[+/+] mouse testes (Supplementary Fig. 4f, g). In order to rule out that the reduction of PRM is not caused by the reduction of sperm count (as indicated in Supplementary Fig. 2c, g), we collected mature sperm in the cauda epididymis, and performed western blots on PRM proteins and found that they were significantly reduced in *Ccer1*[−/−] mice (Fig. 4l). Consistent with this, we found significant histone residues (H2A, H2B, H3 and H4) in sperm from *Ccer1*-deficient mice (Fig. 4k). Next, we also evaluated CMA3 (anthraquinone glycoside chromomycin A3) signal in spermatids, since CMA3 is a fluorescent dye that binds to GC-rich regions of DNA and CMA3 fluorescence in sperm indicates the protamine deficient[17]. We found that testis CMA3 fluorescence signal was significantly elevated in late-stage spermatids in *Ccer1*[−/−] mice compared with wild-type mice. Epididymis sperm staining also confirmed elevated CMA3 signal in *Ccer1*[−/−] sperm (Supplementary Fig. 5a, b). Together, these data support an impaired histone-to-protamine exchange in the *Ccer1*[−/−] mice.

**CCER1 self-assembles to form granule-like condensates**

The change in the steps from 10 to 11 represents a transition in elongating spermatids during the 16-step spermatid development process. We observed that CCER1 forms large granules in the nucleus of spermatids, with concentrated granules evident until step 10, disappearing at steps 11–16 (Figs. 2c, d, and 5a). Similarly, we observed the formation of CCER1 droplet-like condensates in vitro when the GFP-CCER1 plasmid expressing the CCER1 fusion protein was integrated into HEK293T cells (Fig. 5b). The process of GFP-CCER1 condensate formation was also captured by live-cell imaging (Supplementary Movie 1). The CCRE1 protein contains a coiled-coil domain in the

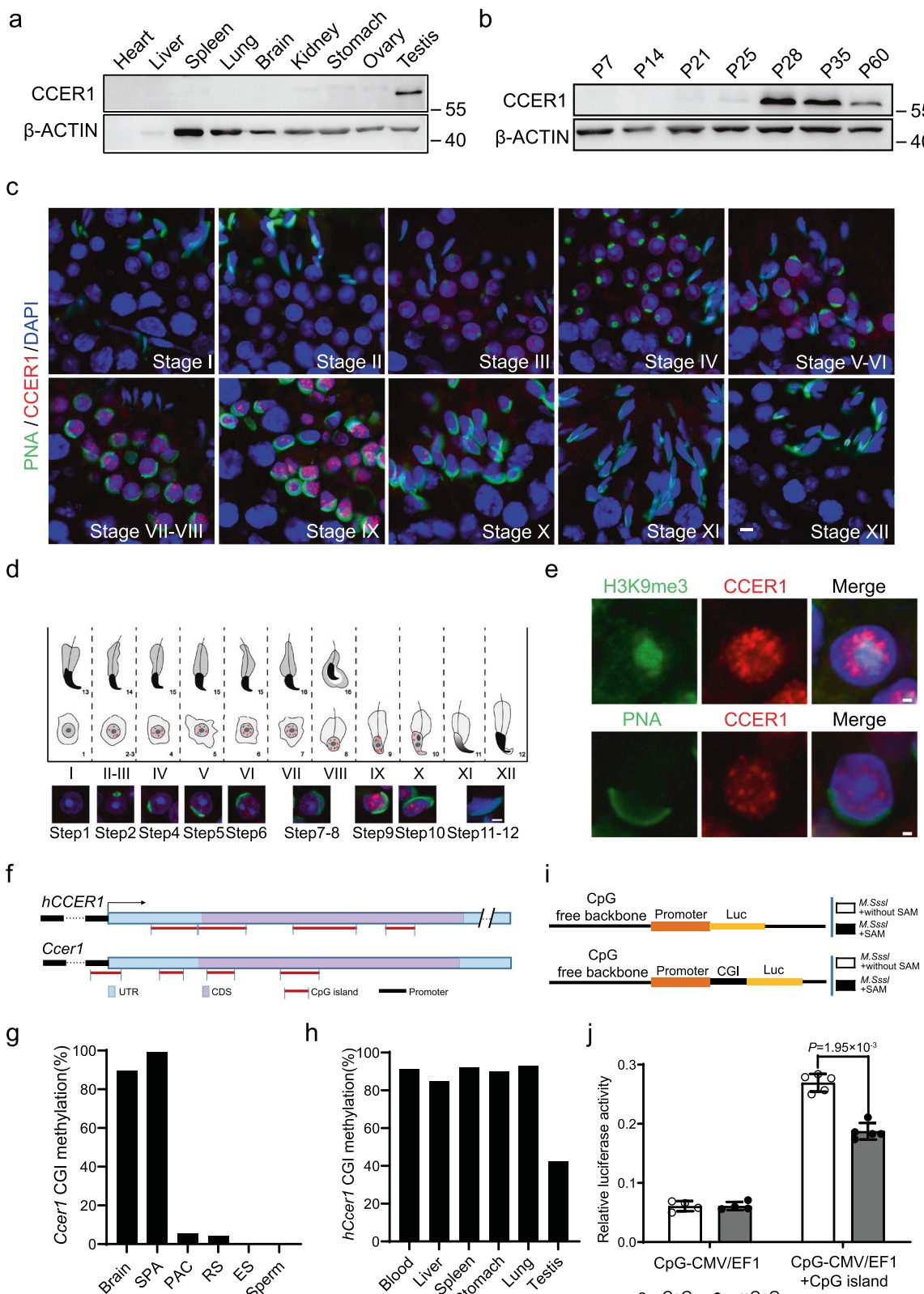

C-terminus, which has been suggested to be a possible dimer- and/or polymer-forming structure. We then explored whether CCER1 undergoes self-interaction and assembly. To this end, we constructed plasmids to generate ectopically expressed mouse CCER1 fusion proteins in HEK293T cells with GFP, mCherry, or Flag protein tags. As determined via co-IP experiments, CCER1 monomers assembled into dimers and polymers through self-interaction (Fig. 5c). Correspondingly,

transiently overexpressed human or mouse CCER1 that was fused to different tag proteins colocalized with each other (Fig. 5d).

We then sought to determine whether the C-terminal coiled-coil domain of CCER1 are self-assembled to form a condensate. We first generated a GFP-CCER1 plasmid expressing a CCER1 fusion protein containing full-length (1–1212 bp) CCER1, CCER1-N (1–873 bp), CCER1-C (1060–1212 bp), CCER1-Δ (874–1059 internal deletion bp), CCER1-CC

**Fig. 2 | Testis-specific expression of *Ccer1* requires demethylation of CGIs.**
**a** Western blot analysis of the CCER1 protein in different tissues of adult mice. β-actin was used as a protein loading control. CCER1 protein is approximately 55KD.
**b** Western blot analysis of the CCER1 protein in mouse testis tissue lysates at different time points (postnatal day, PD) in postnatal development.
**c** Immunofluorescence staining with an anti-CCER1 antibody (red, C-terminal) and PNA (green, peanut agglutinin, a sperm acrosome marker) in adult testes. Nuclear DNA was counterstained with DAPI. Scale bar: 5 μm. **d** Immunofluorescence staining with anti-CCER1 (red) and PNA (green) in stage I–XII in adult testes. Nuclear DNA was counterstained with DAPI. Scale bar: 2 μm. (UP); We drew the schematic during spermatogenesis according to the schematic pattern outlined by Russell L et al.[40] and labelled the expression of CCER1 (red) in the schematic pattern (Down).
**e** Immunofluorescence staining shows that the localization of CCER1 (red) was immiscible with H3k9me3⁺ heterochromatin (green) and DAPI-stained nuclear regions (blue). Scale bar: 1 μm. **f** In silico sequence analysis of the human and mouse

*Ccer1* genes showing the discovery of multiple CpG islands (highlighted in red) in promoter and coding regions. **g** Demethylation of CpG islands was found in late spermatogenic cells (PAC pachytene spermatocytes, RS round spermatid, ES elongating spermatid) in the mouse testis but not in other tissues or spermatogonia (SPA). **h** Methylation of CpG islands in other tissues is much higher than in human testis. **i** Schematic of the human CCER1 CpG island linked to the dual-luciferase (Luc) reporter system. Before transfection, Plasmid CpG-CMV/EF1(upper) and CpG-CMV/EF1+CpG island (bottom) were treated by methyltransferase without S-adenosylmethionine (SAM, the substrates for methyltransferase; *M.SssI+without SAM*) or methyltransferase with SAM (*M.SssI* + SAM), respectively. **j** Effect of CpG islands on CMV/EF1 promoter-driven luciferase expression levels in transfected HEK293T cells. (n = 4 replicate wells for CpG-CMV/EF1 plasmid and n = 5 replicate wells for CpG-CMV/EF1+CpG island plasmid). Two-sided student's *t*-test. Error bars, mean ± SD. P = 1.95 × 10⁻³. Source data are provided as a Source Data file.

(875–1058 bp) or GFP alone to observe the distribution of GFP fusion proteins expressed in HEK293T cells (Fig. 5e). The results showed that only punctate granules were evident in all the plasmid-transfected cells except for the cells carrying the plasmid with the full-length coding sequence of *Ccer1* (Fig. 5f). Nonetheless, the data show that CCER1 interacts with itself and forms substantial condensates.

## Nuclear CCER1 is a phase-separated condensate

It has recently been suggested that the liquid phase mediates the formation of biomolecular condensates associated with different cellular processes. The endogenous and exogenous abundance of these spherical and droplet-like CCER1 aggregates (Figs. 2c, 5a, b, and Supplementary Movie 1) prompted us to hypothesize that the CCER1 protein may undergo a new transition in the nucleus to form a condensate and that CCER1 may exert its biological effects through LLPS.

As intrinsically disordered proteins/regions (IDPs/IDRs) are key molecular drivers that promote LLPS, we examined CCER1 to identify LLPS-driven structure formation. We applied PONDR[18] to analyze the protein sequences of mouse and human CCER1 and found that the globally disordered regions were much longer than the ordered regions. The percentage of CCER1 disordered regions was 70.47% in mice and 66.26% in humans. Five highly conserved disordered regions (defined by a residue number, which was the PONDR score, >0.5, hereafter referred to as IDR 1–5) are located within 403 aa of the full-length mouse CCER1 protein sequence or 406 aa of the full-length human sequence (Fig. 6a, b). Additionally, previous studies have demonstrated some unique polar and charged amino acids enriched in intrinsic IDRs of LLPS-related proteins[6]. We found that CCER1 is a typical glutamate-rich protein, the maximally disordered region of which (IDR5) is located in the C-terminus and is enriched with glutamine. Above all analysis of the CCER1 amino acid sequence strongly indicates an IDR-induced LLPS feature. These IDRs in CCER1 are thought to drive LLPS formation in cells.

Next, we investigated whether CCER1 exhibits LLPS behavior in cells. First, we performed fluorescence recovery after photobleaching (FRAP) to assess whether the mCherry-CCER1 fusion protein condensate is soluble and fluid in cells. As expected, the fluorescence intensity of mCherry-CCER1 recovered rapidly after photobleaching (Fig. 6c, d), indicating that the nuclear CCER1 protein forms droplets in a highly dynamic manner and that these droplets can freely exchange within the nuclear matrix. Second, we performed live-cell imaging and found that when GFP-CCER1 droplets in live HEK293T cells (Supplementary Movie 1) or endogenous CCER1 condensates in nonchemically fixed spermatids were treated with 1,6-hexanediol (a chemical that specifically disrupts LLPS condensates)[19], the CCER1 condensate dissolved rapidly (Fig. 6e, f, and Supplementary Movie 2). Furthermore, via DIC microscopy, we detected clear spherical, droplet-like CCER1 condensates (mCherry-CCER1 fusion protein) in transfected cells

in vitro and testis sections in vivo (Fig. 6g). Liquid phase-separated proteins often form condensates in a specific solution (e.g., 20 μm Tris, 200 mM NaCl, and 1 mM DTT)[20], and we next purified the recombinant CCER1 protein (from yeast) and performed in vitro assays to determine the CCER1 LLPS capacity. The results showed that purified CCER1 indeed formed droplets in the aforementioned solution (Fig. 6h). These results collectively revealed that nuclear CCER1 was a phase-separated condensate in the mouse spermatids.

We then sought to determine whether the defects in h*CCER1* that can be mimicked by mutant variants identified in human patients with NOA (Fig. 1a–c). To this end, we generated ectopically overexpressed mCherry fusion WT and mutant hCCER1 proteins in HEK293T cells (Supplementary Fig. 6a, b). We investigated these mutations in detail and found that the and c.358_388del, 31 bp (p.Cys120fs) mutants led to the degradation of CCER1 proteins and the residual CCER1 was unable to form phase separation; the mutant c.157 C > T (p.Arg53*) and c.534 G > A (p. Trp178*), which induced a truncated hCCER1 protein, lost the ability to form condensates (Supplementary Fig. 6a). All of the abovementioned mutations led to CCER1 LLPS deficiency (Supplementary Fig. 6b).

## CCER1 condensates affects nucleosome epigenetic modifications

As histone writers/readers/erasers are involved in dramatic genome remodeling that rewires the haploid spermatid genome, we next sought to investigate whether there were any epigenetic modification changes in the *Ccer1*⁻/⁻ mouse testes by acetylation, ubiquitination, and methylation, modifications that are required for chromatin conformation, nucleosome stability, and histone–DNA interactions. We first measured the levels of canonical histones (H2A, H2B, H3, and H4) and the linker histone H1 and testis-specific H1 variants. Western blotting revealed that the levels of the linker histone H1.0 and the testis-specific H1 variant H1.6 were decreased in the *Ccer1*⁻/⁻ testis (Fig. 7a), and other histone variants, including H2A/B, H3.1, H3.3, and H4, likely remained intact (Fig. 7b–d). Notably, we found that the ubiquitination levels of H2A/B and the acetylation levels of H3K9, H4K8, K12, and K16 were significantly reduced in the *Ccer1*⁻/⁻ testes (Fig. 7b, d, e, and Supplementary Fig. 7a–c), while the methylation of histone 3 at K4, K9, and K36 likely remained intact (Fig. 7c). Hyperacetylation of histones in particular is a critical step for promoting histone eviction and subsequent TNPs and PRMs incorporation during spermatogenesis; therefore, we verified the hyperacetylation of histones K4 by measuring fluorescence intensity and found that the intensity levels of H4K16ac, at lease (Fig. 7f) was much weaker in the *Ccer1*⁻/⁻ mouse testes than in the *Ccer1*⁺/⁺ mouse testes. Interestingly, the spatiotemporal expression of CCER1 in haploid cells was similarly consistent with the that previously reported for protein CHD5, an epigenetic protein located in heterochromatin that affects histone-to-

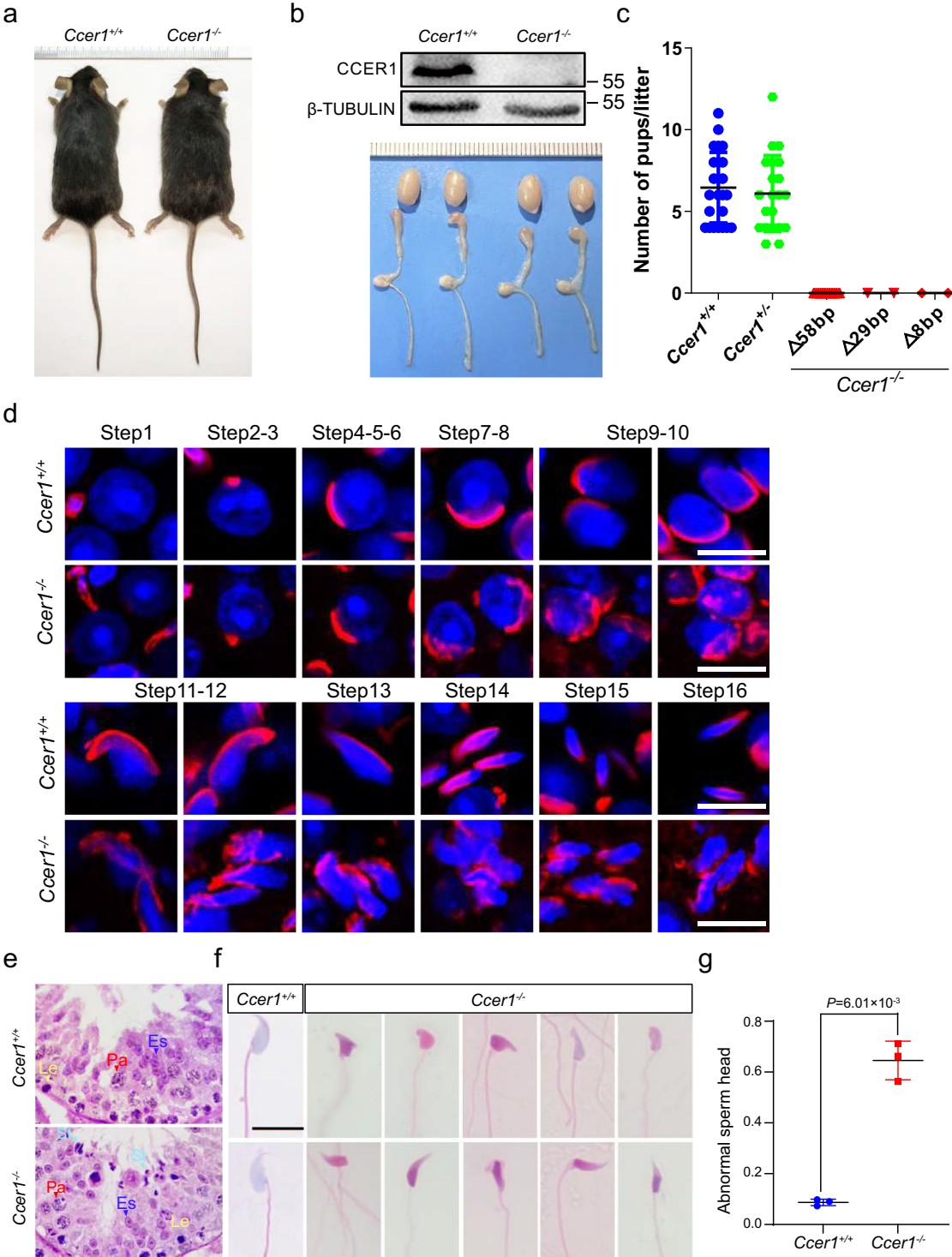

**Fig. 3 | Deletion of mouse CCER1 results in male sterility. a** Body size of *Ccer1⁺/⁺* and CRISPR/Cas9-editing used to generate *Ccer1⁻/⁻* mice. **b** Protein validation of *Ccer1⁺/⁺* and CRISPR/Cas9-mediated deletion of CCER1 and the testis and epididymal morphology in *Ccer1⁻/⁻* mice. The samples derive from the same experiment and that blots were processed in parallel. **c** Breeding experiments with *Ccer1⁻/⁻*, *Ccer1⁺/⁻* and *Ccer1⁺/⁺* males. Mice of each genotype (−58 bp, −29 bp, −8 bp) were crossed with fertile mates. The results (the mean ± SD) were determined using males in cages housing all three *Ccer1⁻/⁻* lines (Homozygous of N_wildtype = 24, N_−58bp = 20, N_−29bp = 2, N_−8bp = 2; and Heterozygous of N_−58bp = 21). **d** PNA staining revealed the morphology associated with the 16-step spermatid development process in the *Ccer1⁻/⁻* testes and *Ccer1⁺/⁺* testes; the results show round-to-elongated spermatids with abnormal sperm heads. Scale bar: 10 μm. **e** Histologically determined morphology of the seminiferous epithelial tissue in the mouse testis revealed a spermiation failure in stage IX (arrow). Scale bar: 20 μm. **f** In addition, compared to those in the *Ccer1⁺/⁺* mice, the spermatozoa in the epididymis of *Ccer1⁻/⁻* mice presented a larger number of malformations. Scale bar: 10 μm. **g** Statistical comparison of the number of abnormal spermatozoa between *Ccer1⁻/⁻* and *Ccer1⁺/⁺* mice (8.70% ± 1.24% and 64.60% ± 7.57%, *n* = 3 for each genotype biological independent mice), Two-sided student's *t*-test. Error bars, mean ± SD. *P* = 6.01 × 10⁻³. Source data are provided as a Source Data file.

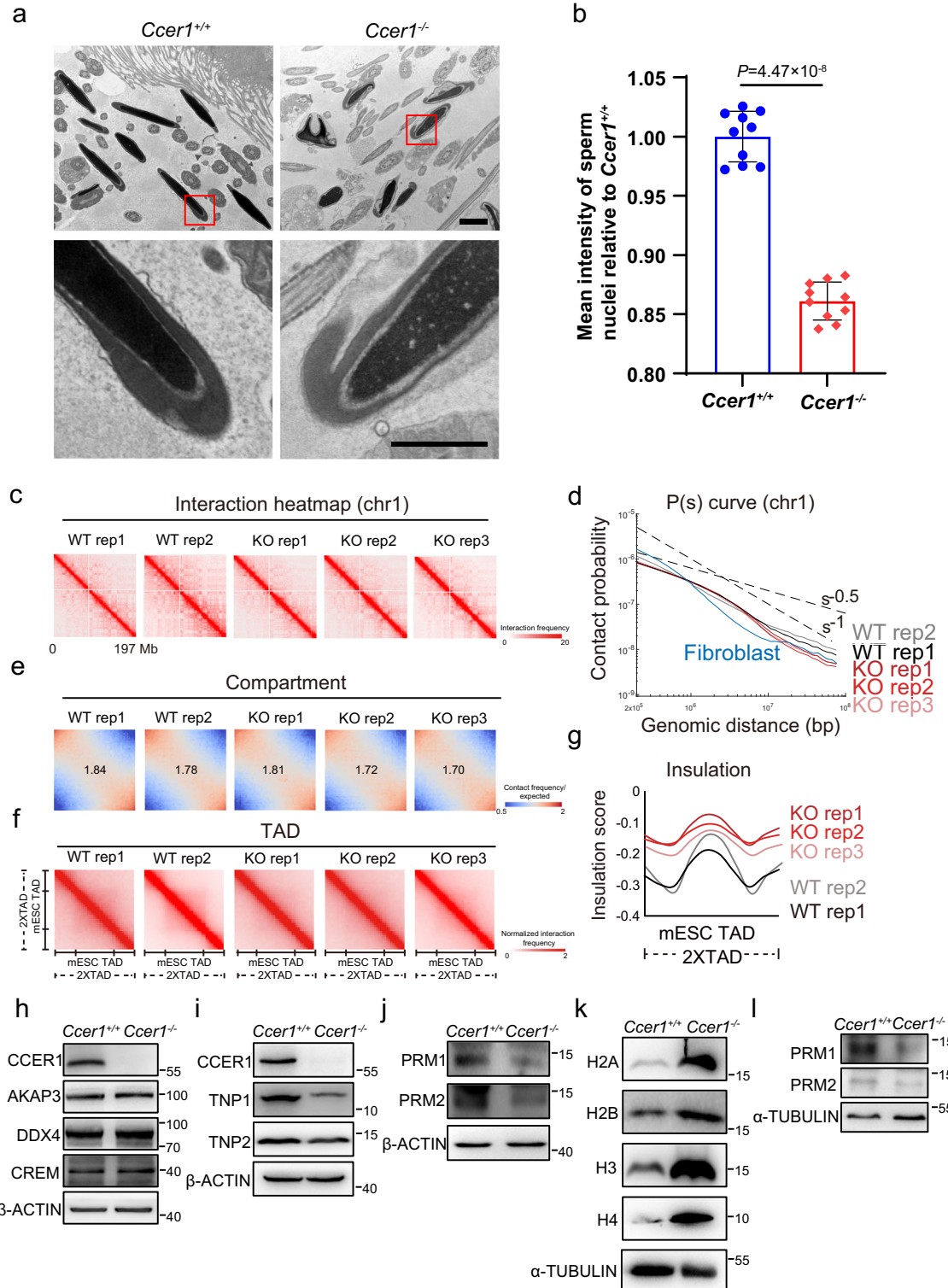

**Fig. 4 | CCER1 mediates *Tnp1/2* and *Prm1/2* transcription and sperm nuclear condensation. a** Representative images of transmission electron micrographs (TEM) showing the defects of DNA condensation in sperm nuclei in *Ccer1*⁻/⁻ mice. Scale bar: 2 μm (upper panel) and 1 μm (lower panel). **b** The grey intensity ratio of *Ccer1*⁻/⁻ vs. *Ccer1*⁺/⁺ sperm was shown (100% ± 2.14% vs. 86.11% ± 1.60%, *n* = 10 replicates). Two-sided student's *t*-test. Error bars, mean ± SD. *P* = 4.47 × 10⁻⁸. Source data are provided as a Source Data file. **c** Heatmaps show the interaction frequency for both *Ccer1*⁺/⁺ and *Ccer1*⁻/⁻ cells (chr1, 500 kb bin, two biological replicates for *Ccer1*⁺/⁺ and three biological replicates for *Ccer1*⁻/⁻ sperms). **d** P(s) curve (100 kb, chr1), which represents the chromatin contact probability relative to genomic distance, is shown for all samples. **e** Saddle plots show the strength of

compartmentalization between *Ccer1*⁺/⁺ and *Ccer1*⁻/⁻ sperms. **f** Heatmaps show the normalized average interaction frequencies around all TADs (defined in mESC) between *Ccer1*⁺/⁺ and *Ccer1*⁻/⁻ replicates. **g** The metaplot shows the insulation score around all TADs for all samples. **h** Western blot analysis of the levels of spermatogenesis-associated proteins in testis. **i** Western blot analysis of the levels of transition proteins in testis. **j** Western blot analysis of the levels of Protamine proteins in testes. **k** Western blot analysis of the levels of H2A, H2B, H3 and H4 in mature sperm. **l** Western blot analysis of the levels of Protamine proteins in mature sperm of *Ccer1*⁺/⁺, and *Ccer1*⁻/⁻ mice. For (**h–l**), the loading control used for quantification and the protein to be compared were derived from the same experiment and that blots were processed in parallel.

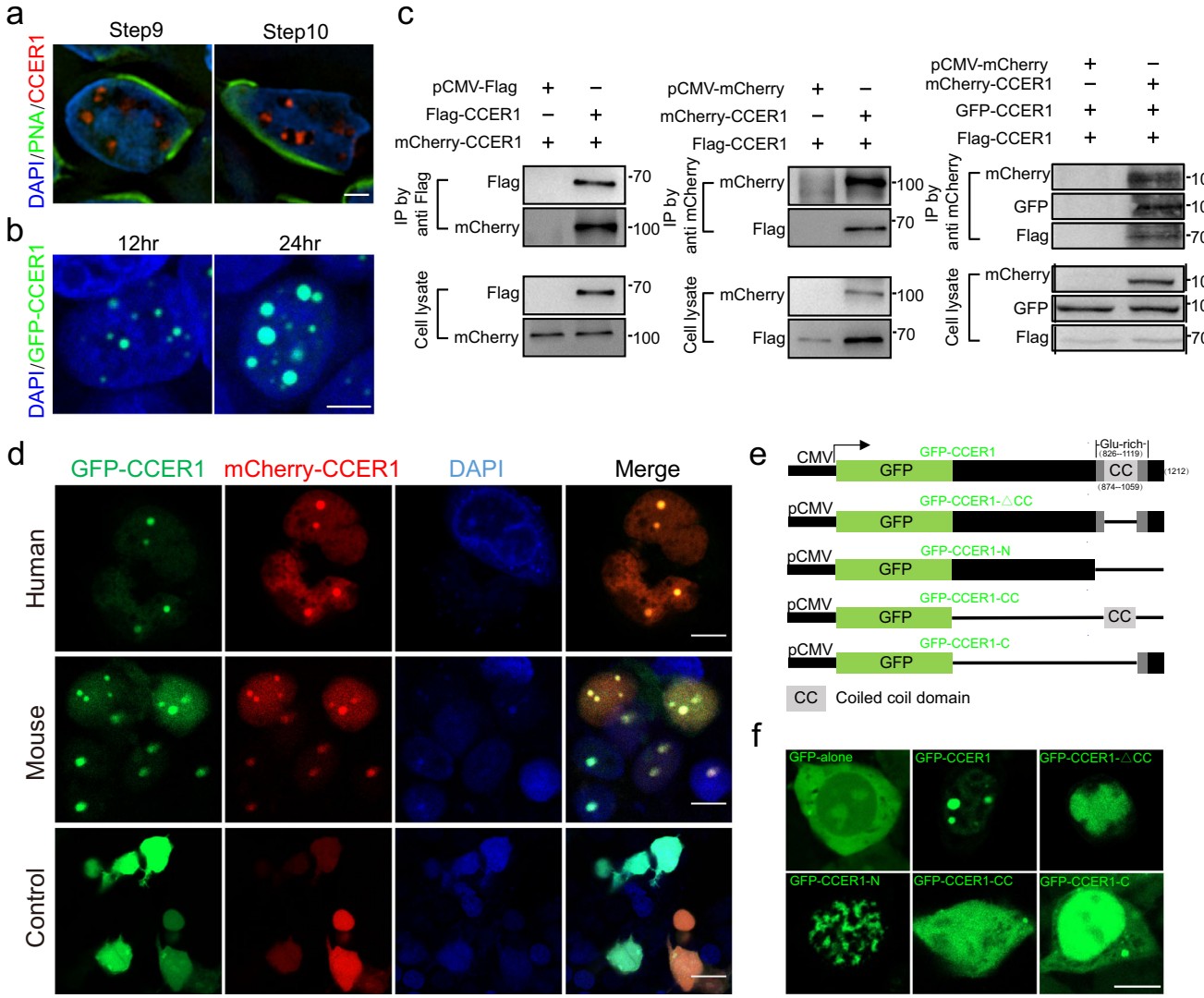

**Fig. 5 | CCER1 self-assembles to form condensates. a** CCER1 forms condensed granules in the DAPI-light chromatin region of elongating spermatids (left, step 9; right, step 10) in vivo. Scale bar: 1 μm. **b** Large condensed granules were found in HEK293T cells transfected with a *Ccer1*-expressing plasmid in vitro. Scale bar: 10 μm. **c** Coimmunoprecipitation of Flag-CCER1 and mCherry-CCER1 from HEK293T cells expressing different CCER1-tagged proteins (Flag, mCherry, or GFP) and immunoblotted with anti-mCherry anti-Flag and anti-GFP antibodies, respectively. **d** Immunofluorescence verification of the colocalization of mCherry-CCER1 and GFP-CCER1 in HEK293T cells transfected with human and mouse s coding sequences. Cells with only GFP transfection were controls. Scale bar: 10 μm. **e** Plasmid constructs expressing full-length CCER1 (1–1212 bp), CCER1-N (1–873 bp), CCER1-C (C terminal, 1060–1212 bp), CCER1-△CC (internal deletion; 874–1059 bp), CCER1-CC (coiled-coil domain, 875–1058 bp) or GFP alone. **f** Granular condensate or punctate distribution in transfected HEK HEK293T cells. Scale bar: 10 μm.

protamine displacement and chromatin remodeling; however, we found that the loss of CCER1 did not affect the CHD5 level or localization (Fig. 7g, h). In addition, we found that not only the total protamine was reduced in *Ccer1⁻/⁻* spermatids (Fig. 4j, l), but also the chromatin-associated protein TNPs and PRMs levels were significantly reduced in the *Ccer1⁻/⁻* spermatids (Fig. 7i, j), strongly indicating that transition proteins failed to replace the histones in the testes of the *Ccer1*-deficient mice. Although there are multiple mechanisms through which LLPS may regulate histone modifications and the transcription of *Tnps* and *Prms* (see discussion), the data taken together suggest that CCER1 liquid-phase condensation is a multifaceted mediator that affects nucleosomal epigenetic modifications in the sperm cell nucleus (Fig. 7k).

Collectively, the results support that nuclear CCER1 forms a phase-separated condensate in the mouse testes, and mutations identified in both human patients and mice affect the phase separation of CCER1 and lead to the loss of CCER1 function and pathogenesis, specifically male infertility.

## Discussion

In this study, we proposed that CCER1, a testis-specific protein that has not been widely studied, is required for spermatogenesis and male fertility. Spermatogenesis is one of the most complex multistage biological processes, with each stage precisely regulated by genes at the transcriptional and posttranscriptional levels. Because DNA methylation is a major mechanism of tissue-specific gene silencing and given that CGIs are discrete CpG-rich regions in 50–70% of human gene promoters[21], we analyzed bona fide CGIs in *Ccer1* and found that they were hypermethylated in normal somatic tissues but not in germ cells, explaining the germline-specific expression of *Ccer1*. Our findings showing that heterozygous human *CCER1* variants are pathogenic in patients with clinical azoospermia were supported by highly significant differences in our large-cohort study. It is worth noting that our data show heterozygous mutations in humans differ in infertility from homozygous mutations in mice, which may be attributed to the following reasons. First, mice, especially those on the B6 background, are less tolerant to deleterious mutations, whereas the complexity of

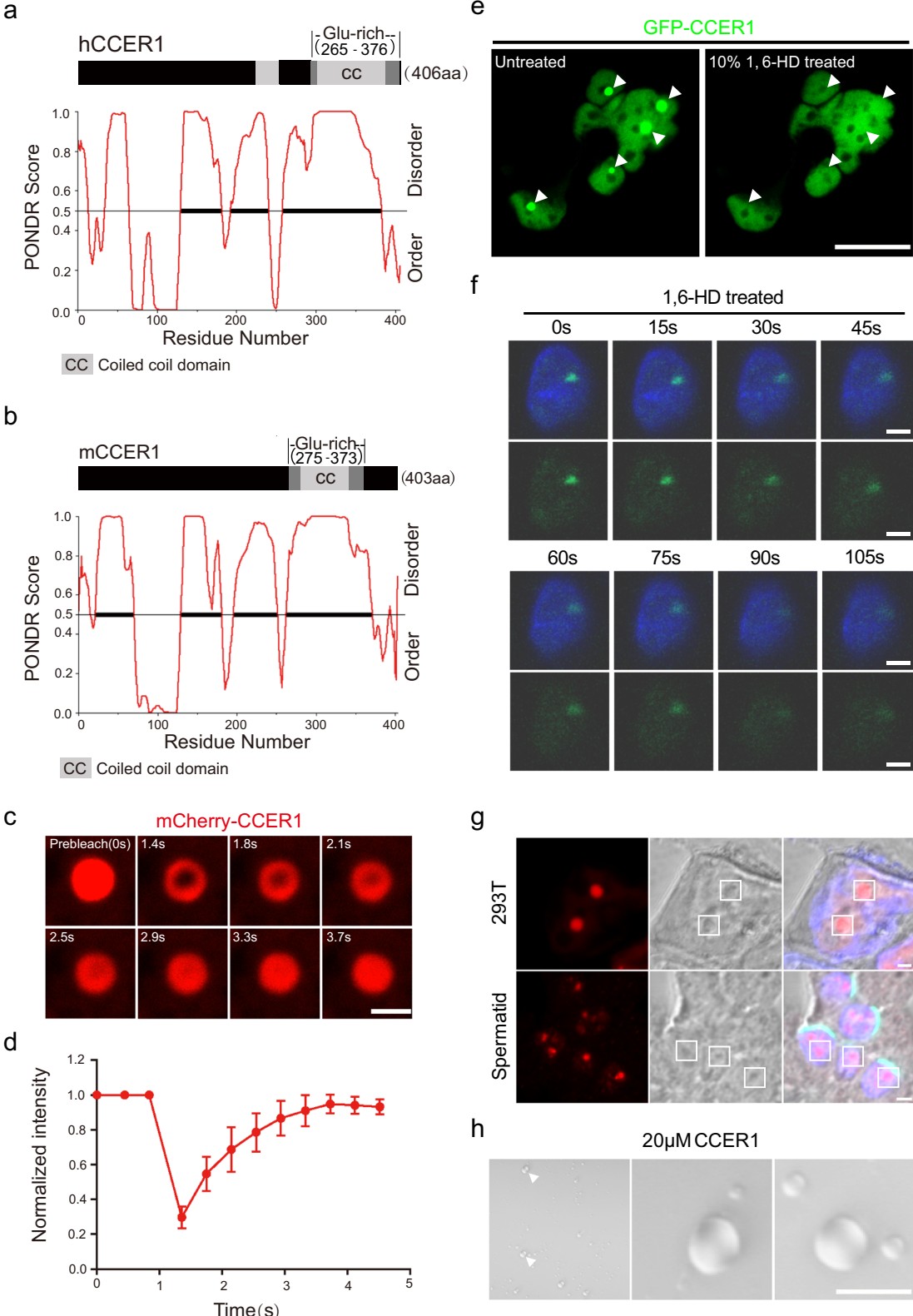

**Fig. 6 | Nuclear CCER1 is a phase-separated condensate protein. a** Disordered regions in the human CCER1 protein sequence, defined by PONDR Score > 0.5, which was based on the number of residues. **b** Disordered region in the mouse CCER1 protein sequence. **c** FRAP analysis of mCherry-CCER1 droplets in transfected cells. Scale bar: 2 μm (**d**). Recovery of normalized fluorescence intensity in the FRAP analysis. *n* = 3 cells. Data are mean ± SD. **e** Treatment with 1,6-hexanediol to disrupt liquid-like CCER1 condensates in living HEK293T cells expressing the GFP-CCER1

fusion protein. Scale bar: 20 μm. **f** Treatment with 1,6-hexanediol to disrupt liquid-like CCER1 condensates in the spermatids of the testis from EGFP-Flag knock-in mice. Scale bar: 2 μm. **g** The CCER1 fusion protein formed droplet-like condensates in HEK293T cells transfected in vitro (up panel); Endogenous droplet-like condensates of CCER1 in spermatids (down panel). Scale bar: 2 μm. **h** Purified recombinant CCER1 protein formed droplets in a specific solution. Scale bar: 10 μm.

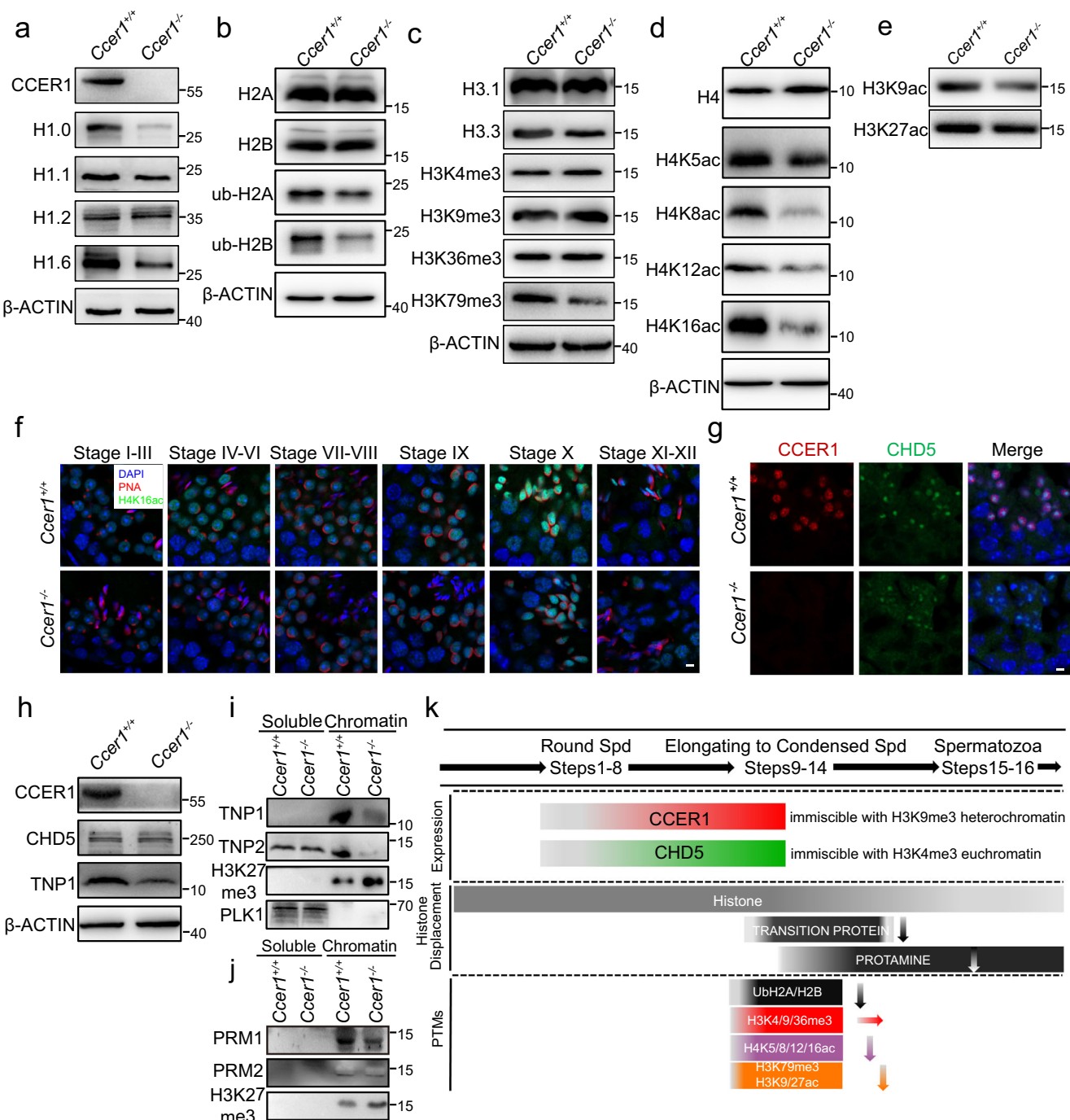

**Fig. 7 | CCER1 liquid-phase condensation affects nucleosome epigenetic modification. a** Representative western blot showing the protein levels of histone 1 variants in the testes of *Ccer1*⁺/⁺ and *Ccer1*⁻/⁻ mice. **b** Representative western blot showing the protein levels of histones 2A and 2B and the ubiquitination of histone H2 variants. **c** Representative western blot showing the protein levels of histone 3 variants and the methylation of histone H3 variants. **d** Representative western blot showing the protein levels of histone 4 and the acetylation of histone H4 lysine residues. **e** Representative western blot showing the acetylation levels of H3K9 and H3K27. **f** Representative immunofluorescence showing the acetylation levels of H4K16 in each stage of seminiferous tubules in the testes of *Ccer1*⁺/⁺ and *Ccer1*⁻/⁻ mice. Scale bar: 5 μm. **g** Immunofluorescence staining showing the localization of CCER1 (red) and CHD5 (green) in mouse testes. Scale bar: 5 μm. **h** Representative

western blots showing the protein levels of CHD5 and TNP1 in the testes of *Ccer1*⁺/⁺ and *Ccer1*⁻/⁻ mice. **i** Chromatin-associated TNP1 and TNP2 were defective in *Ccer1*-deficient mouse spermatid. PLK1 and H3K27me3 were used as loading controls representing the nuclear extract and chromatin-bound fraction, respectively. **j** Chromatin-associated PRM1 and PRM2 were defective in *Ccer1*-deficient mouse spermatid. H3K27me3 as the loading control representing chromatin-bound fraction. **k** Summary of key time points and epigenetic events during the histone-to-protamine transition, when haploid germ cells undergo significant morphological changes and nuclear chromatin reorganization. For (**a**–**e**) and (**h**–**j**), the loading control used for quantification and the protein to be compared were derived from the same experiment and that blots were processed in parallel.

the human genetic background is more tolerant to disease. Tolerance to *CCER1* mutations may contribute to differences in the type and degree of infertility in humans and mice. Second, human genetic variation diseases include not only oligogene mutations that cause multiple lesions, but also the combined effects of multiple gene mutations. Heterozygous NOA patients may also be caused by the combined effect of *CCER1* and other possible lesions.

To further investigate how CCER1 functions in spermatogenesis, we took advantage of in situ Hi-C to illustrate the chromatin state in both wild-type and mutant sperm, and found the density of spermatid nuclei is reduced, the higher-order structure of chromatin is affected, and the interaction of distal chromatin is weakened. What these Hi-C structure changes reflect remains not completely understood. Interestingly, a recent preprint study[22] reported possibility of extracellular chromatin contamination in sperm samples. Based on confocal microscope, the purity of sperm in our study is around 99.29% ± 0.78% in *Ccer1*[+/+] mice and 99.39% ± 0.25% in *Ccer1*[-/-] mice (data not shown), comparable with previous studies[16,23,24]. Nevertheless, we could not exclude the possibility of somatic cell contamination and future studies were warranted to validate these alterations of chromatin organization.

Interestingly, as features of CCER1 include abundant IDRs and dimer or polymer formation, we analyzed CCER1 and found it to be a liquid-phase-separated condensate that is required for chromatin packaging into the sperm head and for fertility because it coordinates the HTP transition in the testis. The HTP transition in testicular haploid cells is a key biological event in which protamine replaces core histone proteins to promote chromatin condensation. During the HTP transition, most somatic histones are first replaced by testis-specific histone variants, followed by the incorporation of transition proteins (TPs) into the spermatid nucleus, and then, protamines (PRMs) replace the TPs. Thus, in late-stage sperm cells, genomes are packaged into highly condensed sperm nuclei[2]. In our study, *Tnp1/2* and *Prm1/2* were downregulated when CCER1 was deleted in haploid cells, and all these proteins are essential for chromatin condensation in spermatids. Many histone variants are expressed during spermatogenesis and regulate the chromatin structure to facilitate histone substitution for protamine. Defects in histone substitutions or modifications can lead to azoospermia, oligospermia, or teratozoospermia, leading to male infertility[25]. The nucleosome is the unit of packaged DNA and contains four typical histones (H2A/2B, H3, and H4) connected through the linker histone H1. In our profiling of histone variants, only the levels of the histone linker H1.0 and testis-specific H1 variant H1.6 were decreased by *Ccer1* deletion in the testis, while other histones, including H2, H3, and H4, likely remained intact. Histone H1 plays a role in linking nucleosomes and stabilizing chromatin structure. Changes in the structure of sperm chromatin may be secondary to a reduction in histone H1. Many posttranslational modifications of histone termini significantly affect chromatin conformation by affecting nucleosome stability and histone–DNA interactions. These modifications have been shown to facilitate the replacement of histones with protamines and mainly include acetylation, ubiquitination, methylation, and phosphorylation[26]. In our study, the acetylation levels in H4 histone (H4K5ac, H4K8ac, H4K12ac, and H4K16ac) and ubiquitination levels in H2 histone (H2A and H2B) in the *Ccer1*-deficient mouse testes, in particular, were lower than those in the *Ccer1* wild type mouse testes. H2A and H2B ubiquitination is prevalent in spermatocytes and elongated spermatids and is required for the recruitment of the acetyltransferase complex to modify H4K16ac and mediate histone removal[3]. H4 hyperacetylation has been shown to be essential for the destabilization and remodeling of nucleosomes and TP incorporation is required for histone-to-protamine substitution during spermatogenesis[5]. Additionally, a previous study showed that acetylation of the histone H4 terminus on lysine residues (K5, K8, K12, and K16) modulated both higher-order chromatin structure assembly and that random hyperacetylation is uniquely important for the formation of chromatin fibers

and is involved in regulating transcriptional activity[27]. The negative effects on H4 hyperacetylation during the HTP transition in *Ccer1*[-/-] mice should be identical. Additionally, other modifications, such as methylation on histone H3, are associated with the transcriptional regulation of *Tnps* and *Prms* gene expression[4,28]. Methylation of H3K4 contributes to the opening of the chromatin configuration, and tri-methylation of H3K9 and H3K27 is associated with the closing of the chromatin configuration, thus balancing the "open" and "closed" chromatin regions during the HTP transition[29]. However, we found no clear changes in H3K4me3, H3K9me3, H3K27me3, and H3K79me3 levels in *Ccer1*-deficient mouse testes, suggesting that mechanisms independent of histone H3 methylation are involved in coordinating the levels of *Tnps* and *Prms*.

Recently, liquid-phase condensation has gained increasing attention, particularly with respect to the nucleus. Nuclear condensates range from micrometer-sized bodies, such as the nucleolus, to sub-micrometer structures, such as transcriptional assemblies, all of which directly interact with and regulate the genome. Separation mechanisms regulate chromatin compartmentalization, chromatin remodeling, and nuclear condensation[30,31]; MeCP2 via LLPS[32,33]. Condensates enhance and disrupt the separation of heterochromatin and euchromatin[33]. Recent research shows that LLPS plays an important role in the ubiquitination modification of nucleosomal H2B components. Lge1 is a scaffold protein that forms a droplet-like structure in the nucleus through phase separation and interacts with the E3 ubiquitin ligase Bre1 through its C-terminal coiled-coil domain to form a Lge1-Bre1 core-shell structure; this structure recruits the ubiquitin-conjugating enzyme and nucleosome strings, which in turn ubiquitinates nucleosome H2[34]. We hypothesized that CCER1 condensates form droplet-like structures similar to those formed by Lge1 to modify histones in spermatid cells.

Multivalent interactions through intrinsically disordered proteins/regions (IDPs/IDRs) that are often enriched with specific polar and charged amino acids, including glycine (G), serine (S), glutamine (Q), proline (P), glutamic acid (E), lysine (K), and arginine (R), are key molecular drivers of liquid condensate assembly[6]. We found that the amino acid sequence of CCER1 has the typical abovementioned characteristics: the C-terminus of CCER1 comprises a coiled-coil domain, which can self-assemble to form higher-order polymers; an overall disordered region that is longer than the ordered region; and an IDR protein, which induces LLPS. More importantly, CCER1 can form a liquid phase in the nucleus and exert its biological function through LLPS, ultimately promoting sperm nuclear condensation during testicular HTP transition. Notably, the three LOF variants we identified in the *CCER1* carried by patients with NOA were located upstream of the coiled-coil domain, leading to premature termination or a frameshift in CCER1 translation. Furthermore, we found that mutant CCER1 proteins with a deleted coiled-coil domain showed no ability to undergo LLPS in vitro. As expected in population studies, the three mutations were absent in genomAD subjects and in our 2713 fertile controls. The combined effect of the above three mutations was significantly associated with an increased risk of NOA, which suggests its pathogenicity of CCER1 loss-of-function mutations. Previously, we have screened the common variants by genome-wide association study in the etiology of NOA in Han Chinese men[35,36], while in this study, we identified *CCER1* rare mutations by Sanger sequencing of the coding region as new idiopathic factors that led to the loss of *CCER1* function and azoospermia.

In sum, our data suggest that *Ccer1* is a gene that causes infertility that mediates nuclear chromatin condensation via LLPS in germ cells. The findings of CCER1 not only hold promise for leading to the elucidation of mechanisms regulating histone replacement with protamine during spermatogenesis and potential treatments for human infertility but also for excellent in vivo model generation for exploring how nuclear phase segregation proteins affect chromatin remodeling.

## Methods

### NOA patient population

All methods and experimental protocols on human subjects were approved by the relevant review of Ethics Committee of Nanjing Medical University. All patients provided written informed consent before taking part in this research. This study included 620 NOA patients recruited from the Nanjing Center of Reproductive Medicine. All infertile male subjects were genetically unrelated Han Chinese men and were selected based on an andrological examination, including examination of their medical history, physical examination, semen analysis, scrotal ultrasound, hormone analysis, karyotyping, and Y chromosome microdeletion screening. Those with a history of cryptorchidism, vascular trauma, orchitis, obstruction of the vas deferens, abnormalities in chromosome number, or microdeletions of the azoospermia factor region on the Y chromosome were excluded from the study. Semen analysis for sperm concentration, motility, and morphology was performed on the basis of the World Health Organization (WHO) criteria (1999). Subjects with NOA showed no detectable sperm in the ejaculate after evaluation of a centrifuged pellet. To differentiate this condition from obstructive azoospermia (OA), only idiopathic azoospermic patients with small and soft testes, normal fructose, and neutral alpha-glucosidase in seminal plasma were included in the study. Those with a history of vasectomy were excluded. To ensure the reliability of the diagnosis, each individual was examined twice, and the absence of spermatozoa from both replicate samples was taken to indicate azoospermia. The 2713 fertile controls were recruited from the adult cohort in Jiangsu Province. A 5-ml sample of whole blood was obtained from each participant as a source of genomic DNA for further Sanger sequencing analysis, and all participants provided written informed consent before participating in this research. The study was conducted in accordance to the criteria set by the Declaration of Helsinki.

### Generation of Ccer1-knockout mice

*Ccer1* was targeted by two sg RNAs as follows: sgRNA1_up: taggGGATCATCTGTGCTGGGCAC; sgRNA1_down: aaacGTGCCCAGCACAGATGATCC; sgRNA2_up: taggGCGTTTGCTGCTGCTCCTGC; and sgRNA2_down: aaacGCAGGAGCAGCAGCAAACGC. Oligos for sgRNA expression plasmids were annealed and cloned into *BsaI* sites of a pUC57-sgRNA plasmid (Cat# 51132, Addgene). Then, sgRNAs were produced and purified using a MEGAshortscript Kit (Cat# AM1354, Ambion) and a MEGAclear Kit (Cat# AM1908, Ambion). Cas9 mRNA was produced and purified using a mMESSAGE MACHINE T7 Ultra Kit (Cat#AM1345, Ambion, USA) and an RNeasy Mini Kit (Cat#74104, Qiagen, Germany). Cas9 mRNA and two sgRNAs were injected into fertilized eggs from C57BL/6 J mice. Embryos were implanted into pseudopregnant C57BL/6 J females according to standard procedures. Finally, we obtained eight founder mice, including three males and four females. Both male and female founders grew normally into adulthood. The chimera founders were backcrossed with C57BL/6 J mice to produce offspring with inherited mutants. All animals were maintained under specific pathogen-free conditions at 20–26 °C and 40–70% humidity in the animal core facility of Nanjing Medical University. All animal procedures were approved by the Institutional Animal Care and Use Committee (IACUC) of Nanjing Medical University (ID: IACUC1912044-3).

### Immunoblotting and immunostaining

We produced an anti-mouse CCER1 antibody with the help from the commercial antibody company Abclonal (Wuhan, China). First, the DNA sequence containing the 179-403aa region of the mouse CCER1 protein was cloned into the pET-32a vector plasmid, and the successful construction of the plasmid was verified by Sanger sequencing. Next, we transformed the pET-32a-CCER1 (179-403aa) plasmid into E. coli (Rosetta strain). A monoclonal strain of E. coli was selected, cultured in LB medium, added with IPTG, and induced at 37 °C for 4 h. Next, cells were harvested by centrifugation at 12,000 g and successful expression was identified by Western blot. Large-scale CCER1 antigen expression was further performed according to the same induction conditions. After purification by affinity column, the final supernatant was used for immunization at a concentration of 1 mg/ml. Japanese white rabbits were immunized four times on the first day, the 12th day, the 26th day and the 40th day respectively, and the animals were sacrificed on the 52nd day to obtain serum antibodies. Furthermore, the serum was subjected to CCER1 antigen affinity purification to obtain purified antibodies.

For western blotting, testes or cells were lysed in a RIPA buffer containing 50 mM Tris-HCl; pH = 7.4, 1% NP-40, 150 mM NaCl, 5 mM EDTA, 0.1% SDS, 0.5% Sodium deoxycholate (Cat# D6750, Sigma Aldrich, USA) and Protease inhibitor cocktail (Cat#11697498001, Roche, Switzerland). Lysates were separated on a 10% SDS-PAGE gel, followed by electrophoresis (Cat#165-8033, Bio-Rad, USA) and electrotransferre (300 mA,100 min) to a 0.22-μm PVDF membrane (Cat# 162-0177, Bio-Rad, USA). Blots were blocked with 5% nonfat milk in Tris-buffered saline containing 0.1% Tween-20 (TBST) for 2 h and then incubated with primary antibodies overnight at 4 °C. Blots were visualized using HRP-conjugated secondary antibodies and ECL reagents (Cat#180-501, Tanon, China).

Because protamines are basic proteins, protamines were then detected by non-denatured native polyacrylamide gel electrophoresis (https://www.bio-rad.com/webroot/web/pdf/lsr/literature/Bulletin_2376.pdf)[37]. Briefly, testis samples were prepared in a buffer (20 mM Tris-HCl; pH = 7.5, 150 mM NaCl, 1% Triton X-100, 5 mM EDTA), containing Protease inhibitor cocktail. Mature spermatozoa were lysed in a buffer (8 M urea, 75 mM NaCl, 50 mM Tris-HCl; pH 8.0) containing protease inhibitor cocktail. The lysates were separated on a 12% native PAGE gel (separating gel: 30% Acr-Bis, 1.5 M Tirs-HCl; pH8.8, 10% APS, TEMED; stacking gel: 30% Acr-Bis, 1 M Tris- HCl; pH6.8, 10%APS, TEMED) followed by electrophoresis and electrotransfer (200 mA, 60 min) to a 0.22-μm PVDF membrane.

For immunostaining, testes or cells were fixed in 4% paraformaldehyde at 4 °C, dehydrated in sucrose, embedded in OCT, and then cut at a 5-μm-thick slice using a cryostat microtome (Cat# CM1900, Lecia, Germany). Sections were exposed to a citrate-based solution (P0083, Beyotime) for antigen retrieval and then incubated with primary antibody and conjugated secondary antibody. Images were captured with a confocal microscope (Model# LSM800, Carl Zeiss AG, Germany). The details of the antibodies were shown in Supplementary Table 1.

### Coimmunoprecipitation

HEK293T cells were transfected with Flag-CCER1, mCherry-CCER1, and GFP-CCER1 plasmids using Powertrans293T reagent (Cat# SX-TR293-10, Sixiang Biological, Shanghai) for 36–48 h before collection. Harvested cells were washed with HBSS and harvested in RIPA buffer (50 mM Tris-HCl, pH = 7.5; 150 mM NaCl; 0.05% SDS; 1 mM EDTA; 1% NP-40; 1 mM DTT; 0.5% sodium deoxycholate; and protease inhibitor cocktail) and sonicated at 4 °C. The supernatants from lysed cells that were centrifuged were incubated with anti-mCherry, anti-Flag, and anti-GFP primary antibodies overnight at 4 °C. Protein A/G beads (Cat# 10002D, Invitrogen) was added to the supernatants and incubated for 4–6 h at 4 °C. After washing the beads with washing buffer, the beads and cell extracts were diluted in SDS loading buffer and subjected to western blot analysis.

### Bisulfite DNA sequencing

DNA was extracted using an Ezup Column Animal Genomic DNA Purification Kit (Cat# B518251, Sangon Biotech). Genomic DNA isolated from each sample was subjected to bisulfite conversion with EZ DNA Methylation-Gold(Cat#D5005, ZYMO research) according to the

manufacturer's instructions. PCR amplification was performed using bisulfite-treated DNA (40–100 ng) as a template. Bisulfite primers were designed against converted DNA sequences. PCR mix (Cat# B600090, Sangon Biotech) with bisulfite-treated genomic DNA in a 50-ml volume was used for amplification. PCR products were subjected to TA cloning according to the manufacturer's instructions. Sanger sequencing was employed to sequence TA clones. For each tissue sample, at least 10 qualified positive clones were selected to evaluate CpG methylation status. The QUMA web-based tool was used to visualize the results (http://quma.cdb.riken.jp/).

### Fluorescence recovery after photobleaching (FRAP) analysis

FRAP experiments were performed with a confocal microscope equipped with a 60× oil immersion objective (LSM 800, Zeiss) at room temperature. Selected regions of mCherry-CCER1 condensates were photobleached at 555 nm, and images were taken every 0.4 s. The intensity was measured on the basis of the mean ROI and further analyzed with GraphPad Prism software.

### In vitro phase separation assay

The in vitro phase separation assay buffer consisted of 20 mM Tris (pH 7.4), 200 mM NaCl, and 1 mM DTT. Purified CCER1 protein was replaced with the in vitro phase separation assay buffer via centrifugal filtration (Millipore, 30 kD). Then, experiments were performed with 0.17-mm microscopy plates (Cellvis). Images were taken with a Zeiss LSM 800 microscope.

### Cloning

The plasmids used in this paper are listed in Supplementary Table 1. Sequences were inserted into plasmids via homologous recombination using a ClonExpress MultiS One Step Cloning Kit (Vazyme Biotech). All sequences cloned into vectors were fully sequenced and then blasted using SnapGene software to confirm the correct insertion of the sequences. All fusion proteins were designed to prevent the generation of frameshift-mutant proteins. For a Dual-Luciferase Reporter assay, a CpG island in the human *CCER1* gene was amplified by high-fidelity PCR (Vazyme Biotech) and then added between the CMV/EF1 promoter and luciferase CDS region in CpG-free luciferase reporter plasmids[14]. NcoI was selected as the restriction site. For the mouse CCER1-truncation protein assay, full-length or truncated fragments of CCER1 were cloned into a pMaxGFP vector (Lonza) and fully sequenced. XhoI was selected as the restriction site for this series of plasmids, and CCER1 was inserted into the C-terminus of the copGFP tag. To mimic human mutants, human full-length *CCER1* or mutated *CCER1* designed on the basis of mutants identified in human patients, were cloned into the mCherry2-C1 vector. XhoI was selected as the restriction site for this series of plasmids, and CCER1 was inserted into the C-terminus of the mCherry tag.

To generate an EGFP-CCER1 construct, full-length mouse *Ccer1* was cloned into pEGFP-C1 vectors. XhoI was selected as the restriction site, and *Ccer1* was inserted into the C-terminus of the EGFP. To generate a Flag-CCER1 construct, full-length mouse *Ccer1* was cloned into a pcDNA3.1-3 × Flag (+) destination plasmid. BamHI was selected as the restriction site. Flag-CCER1, EGFP-CCER1, and mCherry-CCER1 were used in co-IP assays.

### In vitro methylation of plasmid DNA

CpG site-free backbone luciferase (CpG-free luciferase) plasmids were used for an in vitro methylation assay[14]. In vitro methylation of plasmids was realized by using CpG methyltransferase (*M.SssI*) (New England Biolabs; USA) according to the manufacturer's instructions. The reaction system consisted of 10–20 μg of plasmid DNA, *M.SssI* (2.5 U/μg DNA), and 160 μM S-adenosylmethionine (SAM; New England Biolabs). The plasmid DNA and *M.SssI* were incubated for 2 h at 37 °C, and then, SAM equal to the total volume was added, and then, the reaction

system was incubated at 37 °C for an additional 2 h. The unmethylated DNA in the control group was treated in the same way but without the addition of methyltransferase or SAM. Plasmid DNA was purified using an AxyPrep PCR Clean-Up Kit (Axygen; Corning Inc., Corning, USA) and quantified using a Nanodrop spectrophotometer (Thermo Fisher Scientific).

### Dual-luciferase reporter assay

For the dual-luciferase reporter assay, HEK293T cells were seeded at $1.0–1.5 \times 10^5$ cells per 24-well plate. The next day, CpG-free luciferase reporter plasmids were transiently transfected into the HEK293T cells in each well with 200 ng of luciferase reporter vector and 20 ng of Renilla control vector using Lipofectamine reagent (Invitrogen) according to the manufacturer's instructions. Finally, 36–48 h after transfection, the HEK293T cells were harvested to prepare cell lysates for the assay with a dual-luciferase reporter assay system (Promega, Mannheim, Germany). In each transfected-cell group, firefly luciferase activity was normalized to that of Renilla luciferase activity. The results were analyzed with GraphPad Prism.

### Hexanediol treatment

The aliphatic alcohol 1,6-hexanediol (Cat# 240117, Sigma) was added to cells to investigate the properties of CCER1 condensates. For isotonic conditions, 1,6-hexanediol was dissolved in HBSS (Gibco) to a concentration of 10%. Images were collected with a confocal microscope before and after 10% 1,6-hexanediol treatment.

### Hi-C library construction and data processing

Briefly, collected sperm were fixed with 1% formaldehyde for 10 min (RT) and then quenched with glycine (at a final concentration of 0.2 M for 5 min RT). The cells were then washed with PBS, and added to 250 μl of lysis buffer (10 mM pH = 7.4 Tris-HCl, 10 mM NaCl, 0.1 mM EDTA, 0.5% NP-40, proteinase inhibitor), lysed on ice for >15 min, and centrifuged at $2500 \times g$ for 5 min. The pellet was washed once with 500 μl of precooled lysis buffer before centrifuging again at $2500 \times g$ for 5 min. After the supernatant was discarded, adding 50 μl of 0.5% SDS was added to the pellet, which was incubated at 62 °C for 10 min. Subsequently, 145 μl of ddH₂O and 25 μl of 10% Triton X-100 were mixed, added to the detergent and pellet, and incubated at 37 °C for 15 min. Then, 25 μl 10XNEB buffer 2 and 100 U MboI were added to digest chromatin overnight at 37 °C. After digestion, MboI was inactivated at 62 °C for 20 min, and then, the mixture was cooled to RT. Biotins were then added to the DNA in the following reaction system: 20 μl of 0.4 mM biotin-14-dCTP, 0.8 μl of 10 mM dATP/dGTP/dTTP, and 8 μl of 5 U/μl Klenow. The reaction system and DNA were incubated at 37 °C for 1.5 h with rotation. Then, 900 μl of ligation master mix (663 μl of ddH₂O, 120 μl of 10X NEB T4 DNA ligase buffer, 100 μl of 10% Triton X-100, 12 μl of 10 mg/μl BSA, 5 μl of 400 U/μl T4 DNA ligase) was added to the samples and incubated at RT for 4 h to ligate the fragments. After ligation, 50 μl of 20 mg/μl proteinase K and 120 μl of 10% SDS were added to the mixture and incubated at 55 °C for 30 min. Following incubation, 130 μl of 5 M sodium chloride was added and incubated at 68 °C overnight. Finally, the DNA was purified and sheared into 300–500-bp fragments with a Covaris M220 ultrasonicator. The biotin-marked DNA was pulled down with Dynabeads MyOne Streptavidin C1 (Life Technology), and the library was constructed with the beads before sending for Illumina sequencing[1].

Raw reads were processed by HiC-Pro (v 2.9.0) as previously described[1], and the mm9 reference genome was used for mapping and downstream analysis. Valid pairs generated by HiC-Pro were further processed with the hicpro2juicebox.sh script to produce.hic files for visualization[38]. The normalized 40-kb bin matrix generated by HiC-Pro was used to calculate the TAD insulation score[39] as previously described. Cooltools was used for compartment analysis in combination with the valid pairs generated previously. The P(s) curve was calculated with

the normalized 100-kb bin matrix generated by HiC-Pro as previously described[1].

## SEM analysis
For scanning electron microscopy (SEM) assay. the spermatozoa from the cauda epididymis were fixed in 2.5% glutaraldehyde for more than 8 h. After fixed, spermatozoa were dehydrated with ethanol gradient, infiltrated with t-butyl alcohol, and then, samples were freeze-dried (ES2030, Hitachi). Afterward, the spermatozoa were sprayed with gold (E-1010, Hitachi) and scanned (Hitachi SU8010).

## TEM analysis
For transmission electron microscopy (TEM), cauda epididymis was removed from 2-month-old mice and washed with PBS, then put into a 3% glutaraldehyde (pH 7.4) fixing solution immediately. Samples were trimmed to a size of 1 mm*1 mm*3 mm and then washed routinely, post-fixed with 1% osmic acid. Afterward, the samples were washed and dehydrated with an ethanol gradient. Then, dehydrated samples were infiltrated, and embedded with Epon812. The embedded blocks were made into ultrathin sections of 70–100 nm thickness with an LKB-V ultramicrotome (LKB Company, Bromma, Sweden). Finally, the ultrathin sections were counterstained with Pb citrate and uranyl acetate, and then observed with a JEOLL-1200E transmission electron microscope, and recorded with MoradA-G2. Mean gray intensity of sperm nuclei were obtained using Photoshop software (Adobe CS5). Then, relative mean intensity of $Ccer1^{-/-}$ sperm nunclei were calculated by comparing to $Ccer1^{+/+}$.

## Sperm collection, DNA extraction and CMA3 staining
Sperm from $Ccer1^{+/+}$ and $Ccer1^{-/-}$ mice were collected and incubated in Hank's balanced salt solution (Gibco) at 37 °C for 15 min, supernatant was centrifuged at 4 °C 2000 × g for 5 min, and resuspended with 200 µl Hank's balanced salt solution. Next, supernatant was treated with 10 µl Proteinase K (Beyotime) at 45 °C for 2 h. DNA was purified with 210 µl Phenol/Chloroform/Isoamyl alcohol (25:24:1) (Solarbio). DNA in water phase was collected and added with double volume ethanol and 1 µl Glycogen (Beyotime), standing at −20 °C for 2 h before centrifugation at 21130 × g, 24 °C for 15 min. DNA pellet was dissolved with 30 µl nuclease-free water.

Frozen sections (5 µm) of $Ccer1^{+/+}$ and $Ccer1^{-/-}$ mouse testes and mature sperm from the epididymis were applied for CMA3 staining. The slides were fixed with 4% PFA (Paraformaldehyde) at room temperature for 30 min and then blocked for 1 h (3%BSA + 0.1%Tween-20). CMA3 staining was applied with a working concentration at 0.25 mg/ml (Cat: C2659, Sigma Aldrich, USA), dissolved in McIlvane's buffer (0.2 M $Na_2HPO_4$, 0.1 M citric acid, pH 7.0 in 10 mM $MgCl_2$) and incubated at room temperature for 1 h in the dark.

## qRT-PCR and Histological analysis and Sperm analysis
Testes samples were collected from adult $Ccer1^{+/+}$ and $Ccer1^{-/-}$ mice, and RNA extracted using Trizol reagent (Ambion). cDNA was prepared from 1 µg of total RNA through reverse transcription using a Primer-ScriptRT Master Mix (TaKaRa). Diluted cDNA was used for each reaction using SYBR Green Master Mix (Vazyme). A standard 20 µl reaction volume contained forward and reverse primers (200 nM), 2 µl of cDNA, and 10 µl of SYBR Green Master Mix.

Testes and epididymis were fixed in Hartman's fixative (Sigma-Aldrich) for 48 h. Tissues were dehydrated with increasing concentrations of ethanol (70%, 80%, 90% and 100%), cleared in xylene, embedded in paraffin and sectioned (5 µm). Sections were deparaffinized, rehydrated, and stained with Hematoxylin and Eosin (Sigma-Aldrich) or stained with PAS reagent. Sperm from $Ccer1^{+/+}$ and $Ccer1^{-/-}$ mice were extracted and incubated in 90% DMEM (Gibco) + 10% FBS (Gibco) at 37 °C for 15 min, and samples were analyzed using a Computer-Assisted Semen Analyzer (CASA) (IOVS II, Hamilton Thorne, USA).

## Statistics and reproducibility
We conducted a minimum of three independents biological replicates for most of experiments involving histology, immunofluorescence, western blot, immunoprecipitation, and qPCR. For the bisulfite DNA sequencing experiments, the samples in Fig. 2g were derived from a mixture of 6–8 mice, while the human sample number in Fig. 2h is 1. Supplementary Fig. 8 contains all unprocessed scans of the western blots, without any cropping.

## Statistical analysis
The experiments in this work were performed independently at least three times. All quantitative data are presented as the mean ± the standard error of the mean (SEM). Statistical differences between each group were analyzed by two-tailed Student's $t$-tests or rank-sum tests with SPSS software 19.0 (IBM Corporation). A $P$ value < 0.05 was considered to be statistically significant.

## Data availability
The RNA sequencing data and Hi-C datasets reported herein are accessible through the NCBI Gene Expression Omnibus (GEO) with accession number GSE212733. DNA methylation data reported in this study are accessible through Zenodo database (https://zenodo.org/) with code 8077950.

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

## Acknowledgements

We are grateful to Dr. Linyv Lu at Zhejiang University and Dr. Mengcheng Luo at Wuhan University, Dr. P. Jeremy Wang at the University of Pennsylvania, USA, and all fertility research groups (Dr. Y Eugene Xu, X Guo) at SKLRM for their valuable comments and technical help. We thank Dr. Michael Rehli at the Department of Hematology and Oncology; University Hospital; Regensburg, Germany for sharing plasmids, Dr. Liuze Gao for his help in methylation experiments, and Springer nature editing service for help with manuscript grammar checking. We also thank all the members of the Wu, Hu and Xie laboratories for their valuable help. The work was supported by the National Key R&D Program of China (2021YFC2700201 to X.W., 2021YFC2700600 to Z.H., 2019YFA0508901 to W.X.), the Program of Science Fund for Creative Research Groups of the National Natural Science Foundation of China (82221005 to Z.H.), and the National Natural Science Foundation of China (32270897, 32070831, 31872844 to X.W., 31530047, 81830100 to Z.H., 31988101, 31830047, 31725018 to W.X. and 81902836 to Y.G.), and CAMS Innovation Fund for Medical Sciences (2019-I2M-5-064 to Y.Z.).

## Author contributions

X.W., Z.H., W.Xi. conceived and supervised the study. D.Q., S.W., C.C. bred the mice. D.Q., S.W., Y.G., C.C., T.Z. performed most of the experiments except in situ Hi-C experiments. Y.W., K.Z., L.H., and Y.Z. conducted the in situ Hi-C experiments. Y.Z. and W.Xi. performed bioinformatics analysis. Y.G., T.J., and C.W. collected the human samples and performed the genetic data analysis. W.Xu., H.W., and L.L. helped with data analysis and advised on critical knowledge of mouse reproduction. D.Q. drew the schematic of spermatogenesis according to the schematic pattern outlined by Russell L et al. D.Q., Y.G., and X.W. prepared the figures. D.Q., and X.W. wrote the manuscript. The manuscript was reviewed by all authors.

## Competing interests

The authors declare no competing interests.
