## [Peer Review File · Nature Communications]

Phase-separated CCER1 coordinates the histone-to-protamine transition and male fertilityREVIEWER COMMENTS

Reviewer #1 (Remarks to the Author):

The manuscript by Qin et al. entitled "Phase-separated CCER1 coordinates the histone-to-protamine transition and fertility" describes the effects of CCER1 variants in human and murine. In human, missense mutations on one allele, in mice deletion of both alleles render male patients and male mice infertile. The authors demonstrate, that Ccer1 might be regulated by GpC island specific demethylation in germ cells. Further, they show that Ccer1 deficient sperm appears deformed and the whole chromosomal interaction is disturbed. RNA-seq analyses demonstrate that Ccer1 males' sperm display downregulation of TNP and Protamines and an retainment of several Histones in mature sperm. IHC shows, that CCER1 locates to DAPI -dim areas in step9 sperm heads. The protein interacts with each other, to form granule like condensates which are discussed to be associated with different cellular processes. Loss of CCER in testes affects the level of some Histones, ubiquitylated Histones as well as acetylated Histones. In sum, this is an interesting manuscript, which places CCER1 on a list to consider for male infertility. However, there are some shortcomings, which need to be addressed by commenting, amending or performing further experiments.

L64 – most core histones are not replaced by TNPs, but by testes specific histones first – please correct

L76 – the authors must include CCER1 in the introduction to explain the state of knowledge.

L109 – infertility should not be called "disease".

L125 - Fig 1D - better legend needs to be provided. Beta actin staining is highly variable – authors must provide blot with equal loading as shown by beta actin staining.

L133 – do the authors mean "signal" instead of "signaling"?

L136 – and other places – it is not called "sperm deformation" – use the correct term.

L137 – what do the authors mean with the term "DAPI illuminated"?

L128 – 164 there is a general problem with this paragraph. Since the authors have demonstrated before, that CCER1 variants cause infertility, it is not easy to understand, why this paragraph now elaborates on the CpG methylation as means of regulating the expression. Do the authors infer, that problems in methylation also contribute to male infertility. If yes, do they have evidence to support this claim?

Fig 2 improve the legend, explain the labels.

L166 – the authors generate a murine model – here, they need to be more specific about the generation. They need to show the basic analyses such as fertility testing, sperm counts and motility tests. Also, and this appears to be a major point to address – the authors do not present data on Ccer +/- mice. Considering that the patients (as shown in Fig 1, C) harbor one intact allele and carry one variation one would assume that the murine model would also show a pathomorphologic phenotype. The authors must include data of heterozygous animals and discuss differences in phenotype. Further, the authors mention that the patients display NOA – this is quite different from the situation they see in the mice – they must at least comment and explain this apparent difference seen in mice and humans.

L173 – Fig 3 a and b shows two mice, a western blot and three dissected genital tracts, and not as the authors claim " Ccer1-/- mice showed testicular size and body weight ratios comparable to those of their wild-type littermates (Ccer1+/+, Figure 3A and Figure 3B)". The authors must produce a graph displaying body weight and body weight/testicular weight ratio to support their claim.

L176 - how many mice were tested (n) and which generation backcross was used.

L181 – what are the dorsal and ventral fins, the authors write about.

L184 – the authors claim to see head malformations – as these are sections of testes, this is hard to determine at all. What can be seen is a malformation of the Acrosome.

The authors must provide more details in their methods section in order to be able to follow their procedures.

L194- the claim that "loss of CCER1 impedes the development of spermatids" needs to be re-thought.

L190 – the authors show malformation of sperm extracted from the epididymis. This might be a secondary effect, as CCER1 is expressed in earlier stages of spermatogenesis. The authors need to take this into consideration and probably integrate this in their model.

L202 – the authors claim that using TEM, the sperm heads appear less condensed. The pictures provided do not allow to make this point. Here, the authors must provide overview pictures as well as apply a means of quantification of the area and probably percentage of grey and black pixels in the area of interest. Otherwise, this claim seems not supported by the data.

L202 – if there is a problem with condensation, there might be a problem with DNA integrity as well, as much of the respective mutant mousselines published previously showing less densely packed DNA do so. In order to test this, the authors should perform experiments to analyze e.g. 8OHdG levels in testes and epididymis or extract DNA from mature sperm and run it on an agarose gel. If there were DNA damage detected, this must be taken into consideration when interpreting the HiC datasets – damaged DNA might influence the very same parameters measured in Fig 4 B-E and therefore represent a secondary effect.

L229 – Fig 4 F is a pie chart (which can go) and not a scatterplot. The authors probably mean Fig 4 G which is not a scatterplot either.

L231 ff. the analysis of up and downregulated genes is somewhat critical under the aspect, that the authors show in Fig 3E spermiation seems affected. Again, the authors must present data of a thorough basic fertility analysis as suggested above. This would answer the question, whether sperm count is altered. Lower sperm count would explain the reduced levels of Protamine expression.

L240 – In Fig 4J the authors display bands of Prm1 and Prm2 which are run on SDS page gels. This is highly unlikely as the proteins are notorious to be only correctly running on acid urea gels - here the authors must comment and extend on their methods section and probably run acid urea gels for detection of Prm1 and Prm2. Also, could the authors please provide evidence that all bands produced refer to the very same b-actin shown in this figure.

L242 – Fig 4 K, b-actin is missing.

L244 - The mentioned decrease in staining intensity cannot be detected. The authors might want to substantiate the claim of impaired histone to protamine exchange by CMA3 staining, which should nicely pick up areas of chromatin, which is not covered by Prm Proteins.

L337 probably typo H3.1, H3.1

Fig 7 H is not mentioned in text

Fig 7 I methods need to be more precise – again PRM levels should be determined using AU gels.

Reviewer #2 (Remarks to the Author):

In this manuscript, the authors screened patients with NOA (non-obstructive azoospermia) and identified CCER1 as a potential risk gene. Using knockout mouse model and in vitro system, they showed that CCER1 was required for male fertility and depletion of CCER1 impaired histone-to-protamine exchange during spermiogenesis. Moreover, they found that CCER1 was able to form condensates by phase separation in the nucleus of spermatids. This study revealed a new risk factor for male infertility and linked the role of phase separation with histone-to-protamine exchange. Overall, I found the manuscript to be quite well done and worthy of publication in Nature Communications after a bit of minor tweaking and some control experiments to make the story even better.

Major point:

1. Although CCER1 was demonstrated to be essential for male fertility, the causality between male

infertility and phase separation deficiency of CCER1 was not established based on present data. Thus, the function of CCER1 phase separation in histone-to-protamine exchange or spermiogenesis is seemingly overstated.

2.H&E staining of epididymis sections and CASA assays should be performed to examine the effect of Ccer1 knockout on sperm production in mice.

3.Figure 4A, please show a larger field at lower magnification and not just selected images.

4.In Figure 4B-E, the authors showed that chromatin condensation in Ccer1-null spermatids was slightly lower based on the interaction frequency for chromosome 1. Is this change significant? How about other chromosomes?

5.For RNA-seq data in Figure 4, the variation in control groups is relatively high. The authors should check the data quality and consistency between and within different groups. Also, RT-qPCR should be performed to further confirm target gene expression.

6.Nucleosome is composed of linker and octamer histones. Interestingly, Figure 7A-E revealed that the levels of octamer histones were barely affected, while linker histones were obviously decreased in Ccer1-null testes. These points should be discussed.

Minor point:

1.More details about experimental samples and conditions should be provided in figure legends or methods.

2.In Figure 1D, the input should be normalized.

3.Please label each construct in Figure 5E as that in Figure 5F.

4.Please polish the English language. There are number of small grammatical errors that should be corrected, particularly regarding terminology use. For an example, "postmeiotic spermatid differentiation" rather than "postmeiotic sperm differentiation".

Reviewer #3 (Remarks to the Author):

The work of Qin et al is devoted to the study of the role of the CCER1 gene in the maturation of spermatozoa. The authors show that CCER1 forms LLPS condensates at certain stages of spermatozoa differentiation and knockout of this gene in mice results in male sterility. I highly appreciate this work and think that it will be of interest to a wide range of researchers.

However, I have some questions about the manuscript.

1) I have not found information about generation of antibodies in Materials and Method section.

2) I am confused with the sizes of the proteins shown in Figure 1D.

Unfortunately, there is no legend in the text for Figure 1 D.

If the mCherry2 protein is 26kDa and the CCER1 protein is 46kDa, then I would expect the fusion protein to be 72kDa, but on figure 1 D mCherry-hCCER1 is aprox. 100 kDa.

If 100 kDa is a correct size then mutation c.534G>A will shorten protein on 26kDa and expected size will be aprox. 75 kDa. But arrow on Fig 1D points to band 55 kDa.

Authors should provide more detailed explanation of this experiment and explain this size discrepancy.

3) I have had experience with Hi-C on sperm and know that spermatozoa are a very inconvenient cell type for conducting Hi-C, and it is not easy to obtain several libraries of comparable quality. It is clearly seen on the P(s) plot that the wild-type replicas are very different. Could this be due to the different quality of the libraries?

Authors should provide standard library quality metrics similar to those recommended by Encode (Data Production and Processing Standard of the Hi-C Mapping Center

https://www.encodeproject.org/documents/75926e4b-77aa-4959-8ca7-87efc39d79/@@download/attachment/comp_doc_7july2018_final.pdf).

4) It can be seen from the P(s) plot that one can expect a difference in the frequency of contacts at a distance of about 5 Mb. Unfortunately, this range is not visible on the given maps for chromosome 1 (Fig 4B). The authors should add Hi-C maps on which one can appreciate the characteristic structure of TADs and this potential difference in contacts. To do this, authors can show an arbitrary region of

the genome with a size of 40 Mb.

5) «To 218 summarize, these results support the idea that chromatin in Ccer1 mutant sperm 219 underwent condensation at a lower rate»

I have a comment on the interpretation of the P(s) plot. My impression is that the authors interpret the observed decrease in the probability of long-range contacts as confirmation of the less dense chromatin packing observed on TEM. However, the change in P(s) curves cannot be interpreted as a change in chromatin condensation, since the contacts matrix is normalized, therefore, a decrease in distant contacts necessarily leads to an increase in contacts at other distances. A more accurate interpretation is that regions differing in P(s) plot have different principles of chromatin folding.

6) It would be very informative to add a typical cell such as ES cells to P(s) plot.

7) Authors should also share hic files for visualization Hi-C maps.

8) A fast analysis of publications on CCER1 gene shows their almost complete absence. Therefore, I really want the authors to add a minimal evolutionary analysis to the article. When did this gene originate? Why doesn't it have introns? Did it come from a pseudogene?

9) The authors suggest that the variants found in patients lead to azoospermia even in the heterozygous variant, that is, these are autosomal dominant variants. However, in mouse models only homozygous animals are used. Are there fertility problems in heterozygous males? The authors should add this information.

10) In addition, I would like to see an explanation of why having a normal copy of the gene will not save people from azoospermia? For example, the hypothesis that the presence of a truncated peptide does not allow the formation of an LLPS condensate can be easily verified if cells are cotransfected with wt plasmids and a truncated protein. Will condensates similar to those in Figure 5D be formed in such an experiment?

Reviewer 1:

Comments for the author:

The manuscript by Qin et al. entitled “Phase-separated CCER1 coordinates the histone-to-protamine transition and fertility” describes the effects of CCER1 variants in human and murine. In human, missense mutations on one allele, in mice deletion of both alleles render male patients and male mice infertile. The authors demonstrate, that Ccer1 might be regulated by GpC island specific demethylation in germ cells. Further, they show that Ccer1 deficient sperm appears deformed and the whole chromosomal interaction is disturbed. RNA-seq analyses demonstrate that Ccer1 males’ sperm display downregulation of TNP and Protamines and an retainment of several Histones in mature sperm. IHC shows, that CCER1 locates to DAPI -dim areas in step9 sperm heads. The protein interacts with each other, to form granule like condensates which are discussed to be associated with different cellular processes. Loss of CCER in testes affects the level of some Histones, ubiquitylated Histones as well as acetylated Histones. In sum, this is an interesting manuscript, which places CCER1 on a list to consider for male infertility. However, there are some shortcomings, which need to be addressed by commenting, amending or performing further experiments.

1) L64 – most core histones are not replaced by TNPs, but by testes specific histones first – please correct

Response: We thank to the reviewer for pointing out this error. The words had been changed as “most core histones are initially replaced by testes specific histones, and then transition proteins, followed by protamine proteins” (page 3, line 65-66).

2) L76 – the authors must include CCER1 in the introduction to explain the state of knowledge.

Response: According to the suggestion, we have included CCER1 information in the introduction and explain the state of knowledge regarding CCER1 (page 5, line 105-111).

3) L109 – infertility should not be called “disease”.

Response: We appreciate the comments from the reviewer. We have deleted this word.

4) L125 - Fig 1D - better legend needs to be provided. Beta actin staining is highly variable – authors must provide blot with equal loading as shown by beta actin staining.

Response: We thank the reviewer for the comment, which was also pointed out by other reviewers. We have regenerated WB with better protein loading as shown in the β -actin lane. We have also revised the legend of Fig. 1D (page 38, line 953-959), accordingly.

5) L133 – do the authors mean “signal” instead of “signaling”?

Response: We have corrected the word “signaling” to “signal” (L141, L144).

6) L136 – and other places – it is not called “sperm deformation” – use the correct term.

Response: We have revised the description in the text to read "spermatid development".

7) L137 – what do the authors mean with the term “DAPI illuminated”?

Response: We have revised the description of the term “DAPI illuminated”? to “intense DAPI staining”.

8) L128 – 164 there is a general problem with this paragraph. Since the authors have demonstrated before, that CCER1 variants cause infertility, it is not easy to understand, why this paragraph now elaborates on the CpG methylation as means of regulating the expression. Do the authors infer, that problems in methylation also contribute to male infertility. If yes, do they have evidence to support this claim?

Response: We thank the reviewer for this comment. The hypothesis of this paragraph is not to look for evidence that methylation causes male infertility (something we don't know), but to find out why Ccer1 is a testis-specifically expressed gene and not expressed in other tissues. DNA methylation of CpG islands play an important role in the regulation of tissue-specific expression of genes (PMID: 22387149; PMID: 30262495; PMID: 18194566). Therefore, we explored the CpG island methylation of the Ccer1 gene to explain the spatiotemporal specific expression of CCER1, rather than looking for evidence of the effect of methylation on male infertility. To avoid misinterpreting the purpose of this study, we adjusted the relevant sentences (Page 6, line 148-150). These data are present at Figure 2G and 2H.

9) *Fig 2 improve the legend, explain the labels.*

Response: We appreciate the comments from the reviewer. We **have revised the legend and explained the labels in Figure 2.**

10) *L166 – the authors generate a murine model – here, they need to be more specific about the generation. They need to show the basic analyses such as fertility testing, sperm counts and motility tests. Also, and this appears to be a major point to address – the authors do not present data on Ccer +/- mice. Considering that the patients (as shown in Fig 1, C) harbor one intact allele and carry one variation one would assume that the murine model would also show a pathomorphologic phenotype. The authors must include data of heterozygous animals and discuss differences in phenotype. Further, the authors mention that the patients display NOA – this is quite different from the situation they see in the mice – they must at least comment and explain this apparent difference seen in mice and humans.*

Response: Thanks to the reviewer for pointing out the phenotypic difference of CCER1 mutation between humans and mice. First, we argue that the B6 background in mice is less tolerant to deleterious mutations, whereas the complexity of human genetic backgrounds is more tolerant to disease. Thus, tolerance to disease-causing mutations may contribute to differences in the type and extent of disease in humans and mice. A similar example is the phenotypic difference of SYCP3 mutations in mouse and human infertility (PMID: 27997882). Second, human genetic variation diseases include not only oligogene mutations that cause multiple lesions, but also the combined effects of multiple gene mutations. Due to the characteristics of germline inheritance, homozygous infertility mutations cannot be inherited in the population, so single-gene mutations found in infertility inheritance are likely to be heterozygous. For CCER1, we found that NOA patients carried heterozygous abnormalities, but we could not fully assess whether they had other genetic lesions. Therefore, we believe that the above heterozygous NOA patients may also be caused by the joint effect of CCER1 and other possible lesions. In response to this comment, we have provided additional discussion in the revised manuscript (**page 16, line 403-412**).

We thank the reviewer for this comment on the heterozygous phenotype. The data have provided in **Figure 3C in the revision**. Since Ccer1 heterozygous mice had no abnormalities in body and testicular morphology and fertility, pathological analysis was not performed on the heterozygous mice. **Please refer to page 8, line 183-184**. In response to suggestions for other

additional data, we have supplemented homozygous sperm count and motility data using computer-assisted sperm analysis (Figure S2C, S2D, S2E and legend; page 8, line 200-204).

11) L173 – Fig 3 a and b shows two mice, a western blot and three dissected genital tracts, and not as the authors claim “ Ccer1^{-/-} mice showed testicular size and body weight ratios comparable to those of their wild-type littermates (Ccer1^{+/+}, Figure 3A and Figure 3B)”. The authors must produce a graph displaying body weight and body weight/testicular weight ratio to support their claim.

Response: As suggested, we have provided a graph showing the body/testicular weight ratio (Figure S2B and page 8, line 181).

12) L176 - how many mice were tested (n) and which generation backcross was used.

Response: We understand reviewers may be concerned about off-target effects from Cas9 technology. The mice we used were all over 5 generations, and the statistical results started from N=5. This information is provided in the text and legend (page 8, line 179-181 and page 42, line 997-998).

13) L181 – what are the dorsal and ventral fins, the authors write about

Response: We have revised the description of the term to “ventral surface and dorsal surface”.

14) L184 – the authors claim to see head malformations – as these are sections of testes, this is hard to determine at all. What can be seen is a malformation of the Acrosome. The authors must provide more details in their methods section in order to be able to follow their procedures.

Response: In response to the comment, we have re-presented the results for sperm abnormalities in testis section (Fig. 3D), not only seeing acrosomal malformation, but also nuclear malformation at later steps of spermatid development. In addition, our electron microscopy results also revealed extensive sperm head malformations (Figure 4A and 4B).

15) L194- the claim that “loss of CCER1 impedes the development of spermatids” needs to be re-thought.

Response: We agree with the reviewer. We have revised the term “impedes” to “affects”.

16) L190 – *the authors show malformation of sperm extracted from the epididymis. This might be a secondary effect, as CCER1 is expressed in earlier stages of spermatogenesis. The authors need to take this into consideration and probably integrate this in their model.*

Response: We understand that the reviewer’s concern. As seen in our PNA co-stained testis sections, the nuclei of late spermatids were markedly malformation. Therefore, the decrease in sperm count and motility in the epididymis may be due to secondary effects such as delayed spermatid release in the seminiferous lumen. Deletion of other key genes during spermatid development during spermiogenesis showed the same phenotype as CCER1, such as CHD5 expressed in steps 4-10. CHD5 loss also affects histone-protamine replacement in spermatids, with less mature epididymal spermatozoa but a higher proportion of malformation (PMID: 24818823).

17) L202 – *the authors claim that using TEM, the sperm heads appear less condensed. The pictures provided do not allow to make this point. Here, the authors must provide overview pictures as well as apply a means of quantification of the area and probably percentage of grey and black pixels in the area of interest. Otherwise, this claim seems not supported by the data.*

Response: We appreciate the comments from the reviewer. In response to this valid comment, we provided lower and higher-magnification TEM images respectively, and the statistical analysis of gray values showed that the sperm nucleus density decreased significantly after CCER1 deletion (Figure 4A and 4B; page 9, line 217-218). In addition, our sperm Hi-C results strongly suggest that after CCER1 deletion, the distal interaction of chromatin is weakened, which is consistent with the decrease in nuclear density observed by electron microscopy.

18) L202 – *if there is a problem with condensation, there might be a problem with DNA integrity as well, as much of the respective mutant mousselines published previously showing less densely packed DNA do so. In order to test this, the authors should perform experiments to analyze e.g. 8OHdG levels in testes and epididymis or extract DNA from mature sperm and run it on an agarose gel. If there were DNA damage detected, this must be taken into consideration*

when interpreting the HiC datasets – damaged DNA might influence the very same parameters measured in Fig 4 B-E and therefore represent a secondary effect.

Response: At the request of the reviewer, we extracted DNA from WT and KO sperm and performed agarose gel electrophoresis, and the results suggested that there was no significant change in sperm genome integrity after Ccer1 deletion. There is no obvious fragmented DNA that identical to prm2 knockout (PMID: 27833122) by our electrophoresis.

Figure I: Agarose gel electrophoresis of CCER1 mutant sperm DNA

In response to the second question, we used formaldehyde fixation to make intracellular DNA-protein interaction and maintain the 3D structure of intracellular chromatin, followed by enzymatic cleavage, biotin conjugation, enzymatic ligation, DNA extraction and library construction. In the CCER1 knockout sperm, we did not find obvious fragmented DNA, and we do not think it will cause significant impact on the experimental and analytical procedures of Hi C.

19) L229 – Fig 4 F is a pie chart (which can go) and not a scatterplot. The authors probably mean Fig 4 G which is not a scatterplot either.

Response: Thanks to the reviewer for pointing out these errors, we have corrected the description for Figure 4F (Figure S3C in the revised manuscript) to "pie chart" and the description for Figure 4G (Figure S3D in the revised manuscript) to "volcano chart" (page 10, line 246; and page 56, line 1136).

20) L231 ff. *the analysis of up and downregulated genes is somewhat critical under the aspect, that the authors show in Fig 3E spermiation seems affected. Again, the authors must present data of a thorough basic fertility analysis as suggested above. This would answer the question, whether sperm count is altered. Lower sperm count would explain the reduced levels of Protamine expression.*

Response: Thanks to the reviewer for the comments. We have supplemented homozygous sperm count and motility data using computer-assisted sperm analysis (Fig S2C, S2D and S2E). Both mouse epididymis HE results (Fig S2G) and sperm count showed decreased sperm number. This decline can be caused by a number of reasons, including spermiation failure. In order to rule out that the reduction of PRM is not caused by the reduction of sperm count, we also collected mature sperm in the cauda epididymis, and performed western blots on PRM proteins and found that they were significantly reduced (Figure 4L). Together with the evidences of increased residual histones in mature sperm (Figure 4K), as well as IF staining et al, our results support that loss of CCER1 leads to reduced protamine levels in spermatids. Please refer to the writing at page 10, line 259-270 in the revised manuscript.

21) L240 – *In Fig 4J the authors display bands of Prm1 and Prm2 which are run on SDS page gels. This is highly unlikely as the proteins are notorious to be only correctly running on acid urea gels - here the authors must comment and extend on their methods section and probably run acid urea gels for detection of Prm1 and Prm2. Also, could the authors please provide evidence that all bands produced refer to the very same b-actin shown in this figure.*

Response: We thank the reviewers for all comments and sincerely apologize for the sloppiness of our initial draft. In our study, we used native PAGE for PRM western blot (https://www.bio-rad.com/webroot/web/pdf/lsr/literature/Bulletin_2376.pdf; see PMID: 28955874). In order to avoid misleading, we rearranged the WB data of PRMs and separately explained the detection of

PRMs using native PAGE in the method section (page 22, line 587-589). The rearranged data could be found at Figure 4J, 4L and Figure 7J in the manuscript revision.

22) L242 – Fig 4 K, b-actin is missing.

Response: We used alpha tubulin here as an internal reference for Western blot experiments.

23) L244 - The mentioned decrease in staining intensity cannot be detected. The authors might want to substantiate the claim of impaired histone to protamine exchange by CMA3 staining, which should nicely pick up areas of chromatin, which is not covered by Prm Proteins.

Response: Thanks to the reviewer for this valuable suggestion. Using CMA3 staining (Cat: C2659, Sigma Aldrich, USA), we found that testis CMA3 signal was significantly elevated in late-stage spermatids in *Ccer1* null mice compared with wild-type mice. Epididymis sperm staining also confirmed that CMA3 was elevated in *Ccer1*-null sperm. These results further suggested that loss of CCER1 protein affects spermatid histone-protamine replacement (these results are now in Figure S4A and S4B; page 11, line 264-270).

24) L337 probably typo H3.1, H3.1

Response: Typo has been corrected as H3.3.

25) Fig 7 H is not mentioned in text

Response: We thank the reviewer for pointing this out. We have provided the information in the revised manuscript (page 15, line 377).

26) Fig 7 I methods need to be more precise – again PRM levels should be determined using AU gels.

Response: Again, we appreciate the comments from the reviewer. We have provided additional information regarding PRMs detection in the method, please refer our response to point 21.

Reviewer 2:

Comments for the author:

In this manuscript, the authors screened patients with NOA (non-obstructive azoospermia) and identified CCER1 as a potential risk gene. Using knockout mouse model and in vitro system, they showed that CCER1 was required for male fertility and depletion of CCER1 impaired histone-to-protamine exchange during spermiogenesis. Moreover, they found that CCER1 was able to form condensates by phase separation in the nucleus of spermatids. This study revealed a new risk factor for male infertility and linked the role of phase separation with histone-to-protamine exchange. Overall, I found the manuscript to be quite well done and worthy of publication in Nature Communications after a bit of minor tweaking and some control experiments to make the story even better.

1) Although CCER1 was demonstrated to be essential for male fertility, the causality between male infertility and phase separation deficiency of CCER1 was not established based on present data. Thus, the function of CCER1 phase separation in histone-to-protamine exchange or spermiogenesis is seemingly overstated.

Response: We appreciate all positive comments and fully agree with this point raised by the reviewer. Due to the lack of techniques and methods for interfering with protein phase separation in vivo, and the lack of in vitro cell culture for spermatids, it is difficult to study the causal relationship between CCER1 phase separation and its function. We are currently constructing Flag-GFP-labeled transgenic mice of CCER1 to study CCER1 interacting proteins, etc., so as to continue to explore the mechanism of CCER1 phase separation. By using the word "coordinates" in the title of the manuscript, we have weakened the conclusion of the function of CCER1 phase separation in histone-to-protamine exchange.

2) H&E staining of epididymis sections and CASA assays should be performed to examine the effect of Ccer1 knockout on sperm production in mice.

Response: We thank for this suggestion. We have supplemented H&E staining of epididymis sections and CASA assays as suggested, as well as heterozygous fertility data (**Fig 3C, Figure S2C, S2D, S2E, Figure S2G and legend; page 8, line 200-204**). Please also refer to our response to reviewer 1, point 10.

3) Figure 4A, please show a larger field at lower magnification and not just selected images.

Response: According to the reviewer's request, we added the representative image with lower magnification in Figure 4A.

4) In Figure 4B-E, the authors showed that chromatin condensation in *Ccer1*-null spermatids was slightly lower based on the interaction frequency for chromosome 1. Is this change significant? How about other chromosomes?

Response: We thank the reviewer for this comment. We have generated the P(s) curve for all auto chromosomes (Figure IIA below). We listed chr1-5 here as examples (Figure IIB). P(s) curve reflects that WT and KO sample have different principles of chromatin folding. Our data show that the decrease in KO samples is reproducible between replicates (Figure below). We also calculated the p-value (T-test) for the contact probability (contact frequency) at longest distance as it shows largest difference between WT and KO samples in P(s) curve. They are statistically significant.

Figure II: P(s) for all auto chromosomes (A) and chr1-5 as examples (B). WT rep1, KO rep1 and KO rep2 were parallelly performed, while WT rep2 and KO rep3 were conducted together.

5) For RNA-seq data in Figure 4, the variation in control groups is relatively high. The authors should check the data quality and consistency between and within different groups. Also, RT-qPCR should be performed to further confirm target gene expression.

Response: The RNA-seq data in Figure 4 were obtained from three biological replicates per genotype group, where we tried our best to control for biological variation, and these data were also uploaded to the GEO database (accession number GSE212733) for review by potential readers. But we agree with the reviewer's suggestion, we have done RT-qPCR experiments of differential genes, and the overall trend is pretty consistent with the sequencing results. The RT-qPCR data are now present as Figure S3E, and primer information has been provided in the resource table in the revised manuscript.

6) Nucleosome is composed of linker and octamer histones. Interestingly, Figure 7A-E revealed that the levels of octamer histones were barely affected, while linker histones were obviously decreased in Ccer1-null testes. These points should be discussed.

Response: Thanks to the reviewer for the comment. Histone H1 plays a role in linking nucleosomes and stabilizing chromatin structure. In CCER1-deficient sperm nuclei, the density of sperm nuclei is reduced, the higher-order structure of chromatin is affected, and the interaction of distal chromatin is weakened. These causes may be secondary to a reduction in histone H1. Following the reviewer's suggestion, we have added additional discussion of this situation in the revised manuscript (Page 17, line 432-436).

7) More details about experimental samples and conditions should be provided in figure legends or methods.

Response: Thanks to the reviewer for the comment, we have stated "The experiments in this work were performed independently at least three times" in Statistical analysis. We have added more details regarding to experimental samples and conditions in methods in the revision.

8) In Figure 1D, the input should be normalized.

Response: Thank you for your comment on this question, which was also raised by other reviewers. We now present a new set of WB data as shown in Figure 1D. We have also revised

the legend of Fig. 1D (page 38, line 953-959), accordingly. Please also refer to our responses to other reviewers.

9) *Please label each construct in Figure 5E as that in Figure 5F.*

Response: As suggested, we have labeled each construct in Figure 5E.

10) *Please polish the English language. There are number of small grammatical errors that should be corrected, particularly regarding terminology use. For an example, “postmeiotic spermatid differentiation” rather than “postmeiotic sperm differentiation”.*

Response: We thank the reviewer for this comment. We have corrected these words with "postmeiotic spermatid differentiation" in the Abstract. As stated in the Acknowledgments, we have asked for Springer nature editing service with the help on manuscript grammar checking by native English-speaking editors. We also correct all wrong grammatical errors or terminology pointed by all three reviewers in the revision.

Reviewer 3:

Comments for the author:

The work of Qin et al is devoted to the study of the role of the CCER1 gene in the maturation of spermatozoa. The authors show that CCER1 forms LLPS condensates at certain stages of spermatozoa differentiation and knockout of this gene in mice results in male sterility. I highly appreciate this work and think that it will be of interest to a wide range of researchers. However, I have some questions about the manuscript.

1) I have not found information about generation of antibodies in Materials and Method section.

Response: Thanks to the reviewer for the positive comments above and this suggestion, information on CCER1 antibody preparation is now provided in the Methods section (page 21, line 556-570).

2) I am confused with the sizes of the proteins shown in Figure 1D. Unfortunately, there is no legend in the text for Figure 1 D. If the mCherry2 protein is 26kDa and the CCER1 protein is 46kDa, then I would expect the fusion protein to be 72kDa, but on figure 1 D mCherry-hCCER1 is aprox. 100 kDa. If 100 kDa is a correct size then mutation c.534G>A will shorten protein on 26kDa and expected size will be aprox. 75 kDa. But arrow on Fig 1D points to band 55 kDa. Authors should provide more detailed explanation of this experiment and explain this size discrepancy.

Response: We thank the reviewer for this comment. Although the predicted size of CCER1 protein is 46KD, the actual size on blots we observed is about 55KD (as shown by Figure 2A and 2B, 3B and 4I et, al) and the predicted size of mCherry (detected by the recombinant Anti-mCherry antibody, abcam, Cat#ab167453) is 26KD but actually observed size is about 35KD (as shown by our immunoprecipitation experiments, Figure 5C). Therefore, there is some discrepancy between the band position we observed and the predicted position. We attached a whole IP blot of Figure 5C for the reviewer to check the actually observed size of mCherry-hCCER1 (Figure III). Detailed explanation of this experiment and explain this size discrepancy for Figure 1D is provided in the legend (page 38, line 953-959).

Figure III: A representative IP result shown in Figure 5C indicates the position of Ccer1-mCherery is around 100 KD. we have provided the whole blot here for reference.

3) I have had experience with Hi-C on sperm and know that spermatozoa are a very inconvenient cell type for conducting Hi-C, and it is not easy to obtain several libraries of comparable quality. It is clearly seen on the P(s) plot that the wild-type replicas are very different. Could this be due to the different quality of the libraries? Authors should provide standard library quality metrics similar to those recommended by Encode (Data Production and Processing Standard of the Hi-C Mapping Center).

Response: We thank the reviewer for the comment. We have provided the table and a chart for all required sequencing information in Figure S4C and S4D. Due to the special chromatin state of the sperm, the quality is not as well as it is in somatic cells, but the data quality is acceptable and comparable between 2 WT replicate, we have now included all information in Figure S4C and S4D. As for KO samples, all other parameters are reasonable except for the Intra-fragment rate in 2 KO samples, which is a bit higher than 20% (the other one is less than 20%), which may be due to CCER KO effect. We also included others' sperm data here for comparison (PMID: 28709003 and PMID: 31056445). We think they are comparable. In sum, as for all sperm samples, although the quality is not so good and duplicates is high (thus the intra long range rate is marginal), considering the special chromatin state and the KO impact, we think they are acceptable and can be used for the following analysis. We have now added all 5 samples to the

analysis for reference by the reviewer (Figure S4C and S4D). These data we also put here for the reviewer to check (Figure IV).

Rep1	fq	mapped	unique mapped	valid_interaction	valid_interaction_rmsup	trans_interaction	cis_interaction	cis_LongRange	Intra-fragment	Mapping rate	fraction of "Hi-C contacts"	Intra-fragment ratio	Intra Long Range ratio*	Ligation
mouse_sperm_WT_rep1	43021327	3624485	2330053	20778972	13725837	7133734	6592103	6169989	1916194	85.13%	48.30%	8.22%	26.40%	48.20%
mouse_sperm_WT_rep2	74775454	64793977	4290969	38282466	20309582	11244351	9056231	7907617	3696827	86.65%	51.20%	8.61%	18.36%	41.13%
mouse_sperm_KO_rep1	50921001	44096050	2858180	23142098	17034195	7465898	9568297	8690000	4875579	86.60%	45.45%	18.20%	30.06%	42.80%
mouse_sperm_KO_rep2	82511134	81121772	36961598	25953003	19286906	9649234	9637572	8719154	8821024	74.06%	30.41%	20.52%	23.59%	24.92%
mouse_sperm_KO_rep3	12933398	69483253	37724845	22016907	15874576	7843695	7939893	6831197	8379568	89.53%	35.13%	25.00%	20.87%	23.53%
Ke et al.	47806541	45433716	26894287	24840332	24556803	7431346	17128257	15191487	1817524	95.04%	51.96%	6.76%	56.68%	49.36%
Jung et al.	75000000	73865310	44906774	23020264	20667912	7182165	13485747	11480786	10158737	98.51%	30.69%	22.62%	25.57%	47.41%

Figure IV: Sequencing information for all samples.

4) It can be seen from the $P(s)$ plot that one can expect a difference in the frequency of contacts at a distance of about 5 Mb. Unfortunately, this range is not visible on the given maps for chromosome 1 (Fig 4B). The authors should add Hi-C maps on which one can appreciate the characteristic structure of TADs and this potential difference in contacts. To do this, authors can show an arbitrary region of the genome with a size of 40 Mb.

Response: We thank the reviewer for the comment, we have provided the heatmap for the local regions (see Figure V below) for the reviewer to check. As the reviewer mentioned that the sperm Hi-C is not so convenient cell type for conducting Hi-C; however, we have only generated relatively limited cis-long reads, which may be hard for detecting single TAD. Thus, first of all, we also conducted the differential heatmap analysis for whole chr1 (Figure VA for chr1). According to the differential heatmaps, we could see the lower interaction frequency especially in chromosome-scale longer distance (Figure VA) in KO sperm compared with WT in both replicates. Due to the limited cislong reads, we generated the insulation score of local regions chr1: 110Mb-150Mb (Figure VB), the insulation is weaker in local regions in both replicates.

Figure V: Interaction and differential interaction frequency between WT and *Ccer1* KO sperm in chr1 (A) and local region (B, chr1: 110-150 Mb). WT rep1, KO rep1 and KO rep2 were parallelly performed, while WT rep2 and KO rep3 were conducted together. Insulation score for local region chr1: 110Mb-150Mb is also listed below the interaction heatmaps.

5) «To 218 summarize, these results support the idea that chromatin in *Ccer1* mutant sperm 219 underwent condensation at a lower rate» I have a comment on the interpretation of the $P(s)$ plot. My impression is that the authors interpret the observed decrease in the probability of long-range contacts as confirmation of the less dense chromatin packing observed on TEM. However, the change in $P(s)$ curves cannot be interpreted as a change in chromatin condensation, since the contacts matrix is normalized, therefore, a decrease in distant contacts necessarily leads to an increase in contacts at other distances. A more accurate interpretation is that regions differing in $P(s)$ plot have different principles of chromatin folding.

Response: We thank the reviewer for the comment. We have now changed the statement in the manuscript. We changed the original statement with “The P(s) curve clearly showed decreased interaction frequency in distal regions, and this result was highly reproducible, indicating different principles of chromatin folding” (page 9, 229-230), as suggested.

6) *It would be very informative to add a typical cell such as ES cells to P(s) plot.*

Response: Thanks for the comment. We have added the fibroblast as a control for the reviewer to check (Figure VI below).

Figure VI: P(s) curve for mouse fibroblast and all sperm samples.

7) *Authors should also share hic files for visualization Hi-C maps.*

Response: Thanks for pointing this out. We have already uploaded them as well as RNA-seq data to the GEO database (accession number GSE212733) for review by potential readers.

8) *A fast analysis of publications on CCER1 gene shows their almost complete absence. Therefore, I really want the authors to add a minimal evolutionary analysis to the article. When did this gene originate? Why doesn't it have introns? Did it come from a pseudogene.*

Response: We thank the reviewer for the questions and try to answer your questions to the best of our knowledge. We found 206 homologous genes of CCER1 from the NCBI database and This number keeps increasing in the vertebrates ([https://www.ncbi.nlm.nih.gov/gene/?Term=ortholog_gene_196477\[group\]](https://www.ncbi.nlm.nih.gov/gene/?Term=ortholog_gene_196477[group])). We analyzed the exon number of CCER1 and found that CCER1 is a single-exon gene in mammals (such as

humans, mice, rhesus monkeys, etc.) and primitive mammals such as the platypus. In contrast, in some non-mammals such as birds, amphibians, and turtles, CCER1 has two exons bisecting the CDS region, whereas in reptile lizards and snakes, it is a single exon gene. These suggest that CCER1 may have lost introns during the process of evolution, which may increase the transcription and translation efficiency of CCER1. In addition, no other multi-exon genes highly homologous to the CCER1 protein sequence were found through Blast alignment, suggesting that CCER1 may not be derived from the pseudogene formed by other maternal genes through the retrotransposon system (PMID: 25724209).

9) The authors suggest that the variants found in patients lead to azoospermia even in the heterozygous variant, that is, these are autosomal dominant variants. However, in mouse models only homozygous animals are used. Are there fertility problems in heterozygous males? The authors should add this information.

Response: We thank the reviewer for this comment. The heterozygous fertility data have provided in Figure 3C in the revision. Since Ccer1 heterozygous mice had no abnormalities in body and testicular morphology and fertility, pathological analysis was not performed on the heterozygous mice. Please also refer our response to point 10 raised by reviewer 1 and point 2 raised by reviewer 2.

10) In addition, I would like to see an explanation of why having a normal copy of the gene will not save people from azoospermia? For example, the hypothesis that the presence of a truncated peptide does not allow the formation of an LLPS condensate can be easily verified if cells are cotransfected with wt plasmids and a truncated protein. Will condensates similar to those in Figure 5D be formed in such an experiment?

Response: In response to this comment from the reviewer, we did co-transfection experiments of WT-CCER1 EGFP fusion protein, mCherry-CCER1 WT and mCherry-CCER1 mutant proteins. Our results showed that truncated proteins produced by mCherry-CCER1 mutant plasmids were enriched in wild-type EGFP-CCER1 aggregates. The results suggests that CCER1 mutant proteins may play potential dominant-negative effects on normal CCER1 aggregates (see Figure VII below). We are grateful for the reviewer's suggestion, and we will continue to carry out the study on the phase separation mechanism of CCER1 according to your suggestion.

Figure VII: mCherry-hCCER1 WT or mutated plasmids were co-transfected with GFP-hCCER1 plasmid in HEK 293T cells. Scale bar: 20 μ m.

REVIEWERS' COMMENTS

Reviewer #1 (Remarks to the Author):

The authors have amended the ms according to our (and other reviewers) suggestions. From our side, there are no more comments.

Reviewer #2 (Remarks to the Author):

My major concerns are fully addressed in revised manuscript, the manuscript should be accepted.

Reviewer #3 (Remarks to the Author):

The authors did a good job on the manuscript and took into account all my comments. Nevertheless, during the revision, a work important for this area was published (<https://doi.org/10.1101/2022.12.26.521943>), which calls into question the applicability of the Hi-C method for the analysis of the 3D of sperm chromatin. I highly recommend authors to add it to the Discussion section. Since there are serious risks that Hi-C data obtained on sperm from the epididymis are significantly contaminated with extracellular chromatin and do not reflect the features of the organization of sperm chromatin.

Reviewer #1 (Remarks to the Author):

The authors have amended the ms according to our (and other reviewers) suggestions. From our side, there are no more comments.

Response: We express our sincere appreciation for your efforts in reviewing our work and providing invaluable feedback, which has enabled us to enhance the quality of this project.

Reviewer #2 (Remarks to the Author):

My major concerns are fully addressed in revised manuscript, the manuscript should be accepted.

Response: We express our sincere appreciation for your efforts in reviewing our work and providing invaluable feedback, which has enabled us to enhance the quality of this project.

Reviewer #3 (Remarks to the Author):

The authors did a good job on the manuscript and took into account all my comments. Nevertheless, during the revision, a work important for this area was published (<https://doi.org/10.1101/2022.12.26.521943>), which calls into question the applicability of the Hi-C method for the analysis of the 3D of sperm chromatin. I highly recommend authors to add it to the Discussion section. Since there are serious risks that Hi-C data obtained on sperm from the epididymis are significantly contaminated with extracellular chromatin and do not reflect the features of the organization of sperm chromatin.

Response: Thank you for the reviewer's professional suggestion. We cannot rule out the possibility of somatic cell contamination in mouse sperm. However, after discussing with qiangzong, the first author of the paper mentioned by reviewer (<https://doi.org/10.1101/2022.12.26.521943>), they have not yet found a solution to accurately capture the true chromatin structure of sperm. Their use of DTT and Dnase may even damage the actual structure of mature sperm, making it impossible to reflect its true architecture. This is currently a technical limitation. The paper states that "reliably preserving the architecture of the

compacted sperm head presents significant technical challenges that prevent us from confidently assaying true localization.

In summary, as suggested by the reviewer, we include these comments in the discussion section: “Interestingly, a recent preprint reported possibility of extracellular chromatin contamination in sperm samples²². Based on confocal microscopy, the purity of sperm in our study is around $99.29\% \pm 0.78\%$ in *Ccer1*^{+/+} mice and $99.39\% \pm 0.25\%$ in *Ccer1*^{-/-} mice (Figure I), comparable with previous studies^{23, 24, 25}. Nevertheless, we could not exclude the possibility of somatic cell contamination and future studies were warranted to validate these alterations of chromatin organization.”.

Figure I: DAPI staining of mature sperm heads derived from cauda epididymidis. Scale bar: 20 μ m.